# Embarrassingly Simple Dataset Distillation

**Yunzhen Feng**[*‡]   **Ramakrishna Vedantam**[*]   **Julia Kempe**[*†]
[*]Center for Data Science, New York University
[†]Courant Institue of Mathematical Sciences, New York University
[‡]yf2231@nyu.edu

## Abstract

Dataset distillation extracts a small set of synthetic training samples from a large dataset with the goal of achieving competitive performance on test data when trained on this sample. In this work, we tackle dataset distillation at its core by treating it directly as a bilevel optimization problem. Re-examining the foundational back-propagation through time method, we study the pronounced variance in the gradients, computational burden, and long-term dependencies. We introduce an improved method: Random Truncated Backpropagation Through Time (RaT-BPTT) to address them. RaT-BPTT incorporates a truncation coupled with a random window, effectively stabilizing the gradients and speeding up the optimization while covering long dependencies. This allows us to establish new state-of-the-art for a variety of standard dataset benchmarks. A deeper dive into the nature of distilled data unveils pronounced intercorrelation. In particular, subsets of distilled datasets tend to exhibit much worse performance than directly distilled smaller datasets of the same size. Leveraging RaT-BPTT, we devise a boosting mechanism that generates distilled datasets that contain subsets with near optimal performance across different data budgets.

## 1 Introduction

Learning deep, overparameterized neural networks with stochastic gradient descent, backpropagation and large scale datasets has led to tremendous advances in deep learning. In practice, it is often observed that for a deep learning algorithm to be effective, a vast amount of training samples and numerous training iterations are needed.

In this work, we aim to explore the genuine necessity of vast training data and numerous training steps for achieving high test accuracy. To investigate, we limit the number of samples in the training set to be small (e.g., 1, 5, or 10 images per class) and the number of training steps to be small (e.g., on the order of 300 steps). This leads us to the concept of optimizing a small synthetic dataset, such that neural networks trained on this dataset perform well on the desired target distribution, a problem known as *Dataset Distillation* (Wang et al., 2018).

This is an instance of a bilevel optimization problem (Dempe, 2020) where the output of one optimization problem (in this instance, the learning algorithm trained on the small dataset) is fed into another optimization problem (the generalization error on the target set) which we intend to minimize. In general, this problem is intractable, as the inner loop involves a multi-step computation with a large number of steps. Early works (Wang et al., 2018; Sucholutsky & Schonlau, 2021; Deng & Russakovsky, 2022) directly approached this problem via back-propagation through time (BPTT), unrolling the inner loop for a limited number of steps, before hitting an optimization bottleneck that called for alternative techniques. Later works have made steady progress by replacing the inner loop with closed-form differentiable *surrogates*, like the Neural Tangent Kernel (Nguyen et al., 2021a;b), Neural Features (Zhou et al., 2022) and Gaussian Process (Loo et al., 2022). This approach often requires looping thought a diverse pool of randomly initialized models during the optimization to alleviate the mismatch between the surrogate model and the actual one. Moreover, these approaches are limited to MSE loss; they tend to work better on *wider* models, where the surrogate approximation holds better, but give worse performance on the set of frequently used narrower models. Another line of works has modified the outer loop objective using *proxy training-metrics* like matching the trajectories of the network (Cazenavette et al., 2022) or the gradients during training (Zhao & Bilen,

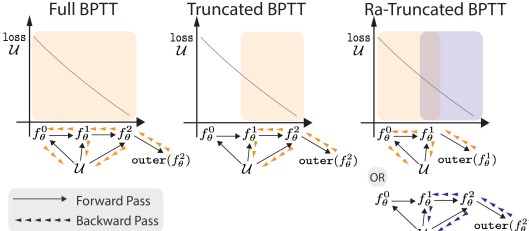

Figure 1: **Illustration of bilevel optimization** of the `outer` loss when training for 3 steps. We show Full Backpropagation Through Time (BPTT) (left), Truncated BPTT (middle) and our proposed Randomized Truncated BPTT (right) (RaT-BPTT). RaT-BPTT picks a window in the learning trajectory (randomly) and tracks the gradients on the training dataset $\mathcal{U}$ for the chosen window, as opposed to T-BPTT that uses a fixed window, and BPTT that uses the entire trajectory.

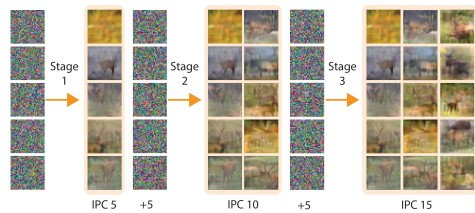

Figure 2: **Boosting Dataset Distillation (Boost-DD).** We start with 5 randomly initialized images per class, distill the dataset into them (Stage 1) yielding five images per class (IPC5), then add five more random images and distill while reducing the learning rate on the first 5 (Stage 2) to yield IPC10, and so on, resulting in a nested dataset of different IPC. Boosting reduces higher order dependencies in the distilled datasets.

2021a). However, these methods either necessitate the storage of different trajectories or are impeded by subpar performance. These observations lead to the question: Does there exist a simple and direct method for dataset distillation?

In this paper, we refine BPTT to address distinct challenges in dataset distillation, and achieve state-of-the-art performance across a vast majority of the CIFAR10, CIFAR100, CUB, TinyImageNet, and ImageNet-1K benchmarks. We start by re-examining BPTT, the go-to method for bi-level optimization problems (Finn et al., 2017b; Lorraine et al., 2020). Notably, the inner problem of dataset distillation presents unique challenges – the pronounced non-convex nature when training a neural network from scratch on the distilled data. One has to use long unrolling of BPTT to encapsulate the long dependencies inherent in the inner optimization. However, this results in BPTT suffering from slow optimization and huge memory demands, a consequence of backpropagating through all intermediate steps. This is further complicated by considerable instability in meta-gradients, emerging from the multiplication of Hessian matrices during long unrolling. Therefore, the performance is limited.

To address these challenges, we integrate the concepts of randomization and truncation with BPTT, leading to the Random Truncated Backpropagation Through Time (RaT-BPTT) method. The refined approach unrolls within a randomly anchored smaller fixed-size window along the training trajectory and aggregates gradients within that window (see Figure 1 for a cartoon illustration). The random window design ensures that the RaT-BPTT gradient serves as a random subsample of the full BPTT gradient, covering the entire trajectory, while the truncated window design enhances gradient stability and alleviates memory burden. Consequently, RaT-BPTT provides expedited training and superior performance compared to BPTT.

Overall, our method is *embarrassingly* simple – we show that a careful analysis and modification of backpropagation lead to results exceeding the current state-of-the-art, without resorting to various approximations, a pool of models in the optimization, or additional heuristics. Since our approach does not depend on large-width approximations, it works for any architecture, in particular commonly used narrower models, for which methods that use inner-loop approximations perform less well. Moreover, our method can be seamlessly combined with prior methods on dataset re-parameterization (Deng & Russakovsky, 2022), leading to further improvements. To our knowledge, we are the first to introduce *truncated* backpropagation through time (Shaban et al., 2019) to the dataset distillation setting, and to combine it with *random* positioning of the unrolling window.

Having established the strength of our method, we proceed to dissect the structure of the learned datasets to catalyze further progress. In particular we address the observation, already made in prior work (e.g. Nguyen et al. (2021b)), that distilled data seems to show a large degree of *intercorrelation*. When training on a subset of distilled data, for instance 10 images per class extracted from a 50-image per class distilled dataset we observe a large degradation in test accuracy: the resulting dataset

performs much worse than if it were distilled from scratch; even worse than training on a dataset of random train images of the same size! This property makes distilled data less versatile since for each desired dataset size we need to distill from scratch. To produce datasets that contain high performing subsamples, we propose *Boost-DD*, a boosting algorithm that produces a nested dataset without these higher-order correlations and only marginal performance loss. It works as a plug-and-play for essentially every existing gradient-based distillation algorithm (see Figure 6 for an illustration). To further our understanding of the learned dataset, we discuss the role of intercorrelation, as well as what information is captured in the distilled data through the lens of hardness metrics.

Overall, our contributions are as follows:

- **RaT-BPTT algorithm:** We propose RaT-BPTT by integrating truncation and randomization with BPTT, and achieve state of the art performance on various dataset distillation benchmarks. RaT-BPTT can be combined with data parametrization methods, leading to further improvements.
- **Boosting:** We propse a boosting-approach to dataset distillation (Boost-DD) that generates a modular synthetic dataset that contains nested high-performance subsets for various budgets.

This paper is structured as follows: Section 2 surveys prior work, Section 3 delineates and motivates our algorithm, RaT-BPTT, Section 4 presents experimental results and compares to prior art, and Section 5 details and evaluates our boosting algorithm. In Section 6 we summarize and discuss bottlenecks to further improvements.

## 2 BACKGROUND AND RELATED WORK

*Dataset Distillation*, introduced by Wang et al. (2018), aims to condense a given dataset into a small synthetic version. When neural networks are trained on this distilled version, they achieve good performance on the original distribution. Dataset distillation shares many characteristics with *coreset selection* Jubran et al. (2019), which finds representative samples from the training set to still accurately represent the full dataset on downstream tasks. However, since dataset distillation generates synthetic samples, it is not limited to the set of images and labels given by the dataset and has the benefit of using continuous gradient-based optimization techniques rather than combinatorial methods, providing added flexibility and performance. Both coresets and distilled datasets have found numerous applications including speeding up model-training Mirzasoleiman et al. (2020), reducing catastrophic forgetting Sangermano et al. (2022); Zhou et al. (2022), federated learning Hu et al. (2022); Song et al. (2022) and neural architecture search Such et al. (2020).

Numerous follow up works have proposed clever strategies to improve upon the original direct bilevel optimization(see Lei & Tao (2023); Sachdeva & McAuley (2023); Yu et al. (2023); Geng et al. (2023) for recent surveys and Cui et al. (2022) for benchmarking). Yet, given the apparent intractability of the core bilevel optimization problem, most works have focused on 1) approximating the function in the inner-loop with more tractable expressions or 2) changing the outer-loop objective.

*Inner-loop surrogates:* The first innovative works Nguyen et al. (2021a;b) tackle inner-loop intractability by approximating the inner network with the Neural Tangent Kernel (NTK) which describes the neural net in the infinite-width limit with suitable initialization (Jacot et al. (2018); Lee et al. (2019); Arora et al. (2019)) and allows for convex optimization, but scales unfavorably. To alleviate the scaling, random feature approximations have been proposed: Loo et al. (2022) leverage a Neural Network Gaussian process (NNGP) to replace the NTK, using MC sampling to approximate the averaged GP. Zhou et al. (2022) propose to use the Gram matrix of the feature extractor as the kernel, equivalent to only training the last layer with MSE loss. A very recent work Loo et al. (2023) assumes that the inner optimization is convex by considering linearized training in the lazy regime and replaces the meta-gradient with implicit gradients, thus achieving most recent state-of-the-art. Yet all of these approaches inevitably face the discrepancies between learning in the lazy regime and feature learning in data-adaptive neural nets (e.g. Ghorbani et al. (2019) and numerous follow ups) and often need to maintain a large model pool. Moreover, inner-loop surrogates, be it NTK, NNGP or random features, tend to show higher performance on *wide* networks, where the approximation holds better, and be less effective for the narrower models used in practice.

*Modified objective:* A great number of interesting works try to replace the elusive test accuracy objective with metrics that match the networks trained on full data and on synthetic data. Zhao & Bilen (2021a) propose to match the gradient between the two networks with cosine similarity,

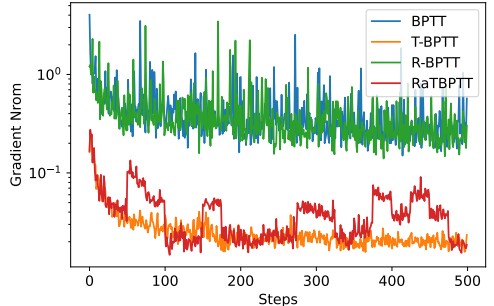

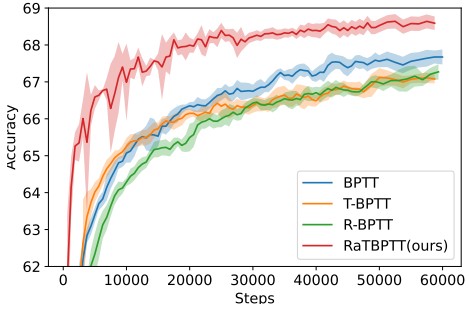

Figure 3: Meta-gradient norm in the first 500 steps. BPTT (unroll 120 steps) have unstable gradients. T-BPTT (unroll 120 steps and backpropagate 40 steps) stabilizes the gradient. For RaT-BPTT, for each epoch (25 batch-update steps) we randomly place the 40-step backpropagation window along the 120 unrolling. CIFAR10, IPC10.

Figure 4: Test Accuracy during distillation with BPTT, T-BPTT, R-BPTT, and our RaT-BPTT. Using random unrolling (R-BPTT) and truncated window (T-BPTT) are both worse than BPTT. Combining them into RaT-BPTT gives the best performance. CIFAR10, IPC10.

with various variations (like differentiable data augmentation (Zhao & Bilen, 2021b) (DSA)) and improvements (Jiang et al., 2022; Lee et al., 2022b). Other works pioneer feature alignment (Wang et al., 2022), matching the training trajectories (MTT, introduced in Cazenavette et al. (2022) and refined in Du et al. (2023); Cui et al. (2023); Zhang et al. (2022b)), and loss-curvature matching (Shin et al., 2023). However, it is unclear how well the modified outer-loop metrics align with the test-loss objective and most of these methods ends up with subpar performance.

## 3 METHODS

In this section, we start by defining the dataset distillation problem to motivate our RaT-BPTT method. Denote the original training set as $\mathcal{D}$ and the distilled set as $\mathcal{U}$. With an initialization $\theta_0$ for the inner-loop learner $\mathcal{A}$, we perform the optimization for $T$ steps to obtain $\theta_T(\mathcal{U})$ with loss $\mathcal{L}(\theta_T(\mathcal{U}), \mathcal{D})$. We add $(\mathcal{U})$ to denote its dependence on $\mathcal{U}$. The dataset distillation problem can be formulated as

$$\min_{\mathcal{U}} \mathcal{L}(\theta_T(\mathcal{U}), \mathcal{D}) \text{ (outer loop)} \quad \text{such that} \quad \theta_T(\mathcal{U}) = \mathcal{A}(\theta_0, \mathcal{U}, T) \text{ (inner loop)} \quad (1)$$

The principal method for tackling bilevel optimization problems is *backpropagation through time (BPTT)* in reverse mode. When the inner-loop learner $\mathcal{A}$ is gradient descent with learning rate $\alpha$, we obtain the meta-gradient with respect to the distilled dataset by leveraging the chain rule:

$$\mathcal{G}_{BPTT} = -\alpha \frac{\partial \mathcal{L}(\theta_T(\mathcal{U}), \mathcal{D})}{\partial \theta} \sum_{i=1}^{T-1} \Pi_{j=i+1}^{T-1} \left[ 1 - \alpha \frac{\partial^2 \mathcal{L}(\theta_j(\mathcal{U}), \mathcal{U})}{\partial \theta^2} \right] \frac{\partial^2 \mathcal{L}(\theta_i(\mathcal{U}), \mathcal{U})}{\partial \theta \partial u} \quad (2)$$

The aforementioned computation reveals that the meta-gradient can be decomposed into $T - 1$ parts. Each part essentially represents a matrix product of the form $\Pi[1 - \alpha H]$ where every $H$ matrix corresponds to a Hessian matrix. Nonetheless, computing the meta-gradient demands the storage of all intermediate states to backpropagate through every unrolling step. This imposes a significant strain on GPU memory resources and diminishes computational efficiency.

To circumvent these challenges, the prevalent strategy is *truncated* BPTT (T-BPTT) method (Williams & Peng, 1990; Puskorius & Feldkamp, 1994), which unrolls the inner loop for the same $T$ steps but only propagates backwards through a smaller window of $M$ steps. In T-BPTT, the gradient is

$$\mathcal{G}_{T-BPTT} = -\alpha \frac{\partial \mathcal{L}(\theta_T(\mathcal{U}), \mathcal{D})}{\partial \theta} \sum_{i=T-M}^{T-1} \Pi_{j=i+1}^{T-1} \left[ 1 - \alpha \frac{\partial^2 \mathcal{L}(\theta_j(\mathcal{U}), \mathcal{U})}{\partial \theta^2} \right] \frac{\partial^2 \mathcal{L}(\theta_i(\mathcal{U}), \mathcal{U})}{\partial \theta \partial u} \quad (3)$$

The distinguishing feature of T-BPTT is its omission of the first $T - M + 1$ terms in the summation; each omitted term is a product of more than $M$ Hessian matrices. Under the assumption that the inner loss function is locally $\alpha-$strongly convex, Shaban et al. (2019) shows that T-BPTT

inherits convergence guarantees. The theoretical result comes from the diminishing contributions of the Hessian products. Strong convexity assumptions endow the Hessian matrices with positive eigenvalues. Consequently, $1 - \alpha H$ will have all eigenvalues smaller than 1, and the product term $\Pi[1 - \alpha H]$ vanishes as the number of factors increases. Therefore, T-BPTT could enjoy a similar performance compared with BPTT but with less memory requirement and faster optimization time.

However, the inner task in our context diverges significantly from the realm of strong convexity. It contains training a neural network from scratch on the current distilled data with random initialization. This problem is intrinsically non-convex with multiple local minima. This beckons the question: how do BPTT and T-BPTT fare empirically?

We visualize the training curve and the norm of meta-gradients through outer-loop optimization steps in Figure 4. The experiment is performed on CIFAR10 with IPC 10. A comparison between BPTT120 and T-BPTT reveals that: 1) The meta-gradients of BPTT manifest significantly greater instability than their T-BPTT counterparts. This observed volatility and norm discrepancy can be attributed to the omitted $T - M + 1$ gradient terms. It underscores the highly non-convex nature of the inner problem, characterized by Hessian matrices with negative eigenvalues. The compounded effects of these negative eigenvalues amplifies the variance from different initializations, creating the unstable gradient behavior. With the gradient stabilized, T-BPTT achieves faster improvement dur-

---

**Algorithm 1** Dataset Distillation with RaT-BPTT. Differences from BPTT are highlighted in purple.

**Input:** Target dataset $\mathcal{D}$. T: total number of unrolling steps. M: truncated window size.

1: Initialize distilled data $\mathcal{U}$ from Gaussian
2: **while** Not converged **do**
3:     Uniformly sample N in $[M, T]$ as the current unrolling length
4:     Sample a batch of data $d \sim \mathcal{D}$
5:     Randomly initialize $\theta_0$ from $p(\theta)$
6:     **for** $n = 0 \rightarrow N - 1$ **do**
7:         If $n == N - M$, start accumulating gradients
8:         Sample a mini-batch of distilled data $u_t \sim \mathcal{U}$
9:         Update network $\theta_{n+1} = \theta_n - \alpha \nabla \ell(u_n; \theta_n)$
10:    **end for**
11:    Compute classification loss $\mathcal{L} = \ell(d, \theta_N)$
12:    Update $\mathcal{U}$ with respect to $\mathcal{L}$.
13: **end while**

---

ing the initial phase. 2) BPTT ends up with higher accuracy than T-BPTT. This indicates that important information from the initial phase is disregarded in T-BPTT — a notable concern given that the rapid optimization of neural networks usually happens during the early stage of the inner loop. The challenge thus is how to harmonize the good generalization performance of BPTT with the computational speedup of T-BPTT.

To this end, we propose the *Random* Truncated BPTT (RaT-BPTT) in Algorithm 1, which randomly places the truncated window along the inner unrolling chain. The gradient of RaT-BPTT is

$$\mathcal{G}_{RaT-BPTT} = -\alpha \frac{\partial \mathcal{L}(\theta_N(\mathcal{U}), \mathcal{D})}{\partial \theta} \sum_{i=N-M}^{N-1} \Pi_{j=i+1}^{N-1} \left[ 1 - \alpha \frac{\partial^2 \mathcal{L}(\theta_j(\mathcal{U}), \mathcal{U})}{\partial \theta^2} \right] \frac{\partial^2 \mathcal{L}(\theta_i(\mathcal{U}), \mathcal{U})}{\partial \theta \partial u} \quad (4)$$

RaT-BPTT varies from BPTT by randomly selecting M consecutive segments from $\mathcal{G}_{BPTT}$ and excluding shared Hessian matrix products. Essentially, RaT-BPTT is a subsample of BPTT, covering the entire learning trajectory, but limits the number of Hessians in the product to less than M. This approach combines T-BPTT's accelerated performance and gradient stabilization. As shown in Figure 4, RaT-BPTT consistently surpasses other methods in the optimization process. Contrastingly, R-BPTT, involving full unrolling over randomly sampled trajectory lengths, yields unstable gradients and inferior performance compared to full BPTT unrolling. Section 4.3 includes an ablation study on the importance of a moving truncated window and the rationale behind random uniform sampling in Algorithm 1. We investigate the stability of meta-gradients using gradient norms as a metric, predicated on the notion that stable and efficient learning should manifest as consistent and decreasing gradient norms throughout training. We further provide instability analysis in Appendix D.3 with another metric, the normalized variance. In light of the significant change of the network within the inner loop, concurrent work Chen et al. (2024) introduces a progressive concept that bears resemblance to our random design approach.

## 4 EXPERIMENTAL RESULTS

In this section, we present an evaluation of our method, RaT-BPTT, comparing it to a range of SOTA methods across multiple benchmark datasets.

**Datasets** We run experiments on four standard datasets, CIFAR-10 (10 classes, $32 \times 32$), CIFAR-100 (100 classes, $32 \times 32$, Krizhevsky et al. (2009)), Caltech Birds 2011 (200 classes, CUB200, $32 \times 32$, Wah et al. (2011)), Tiny-ImageNet (200 classes, $64 \times 64$, Le & Yang (2015) ), ImageNet-1K (1,000 classes, $64 \times 64$, Deng et al. (2009)). We distill datasets with 1, 10, and 50 images per class for the first two datasets, with 1 and 10 images per class for the last two datasets, and 1 and 2 images per class for the big ImageNet-1K dataset.

**Baselines** We compare our methods to the first two lines of works as we discussed in related work (Section 2), including 1) *inner-loop surrogates*: standard BPTT (the non-factorized version of LinBa in (Deng & Russakovsky, 2022)), Neural Tangent Kernel (KIP) (Nguyen et al., 2021b), Random Gaussian Process (RFAD) (Loo et al., 2022), and empirical feature kernel (FRePO) (Zhou et al., 2022), and reparameterized convex implicit gradient (RCIG) (Loo et al., 2023), 2) *Modified objectives*: gradient matching with augmentation (DSA) (Zhao & Bilen, 2021b), distribution matching (DM) (Zhao & Bilen, 2023), trajectory matching (MTT) (Cazenavette et al., 2022), a memory-friendly version of MTT (TESLA) (Cui et al., 2023) and flat trajectory distillation (FTD) (Cui et al., 2023). The works on parametrization (Deng & Russakovsky, 2022; Liu et al., 2022; Kim et al., 2022) are complementary to our optimization framework and can be combined with RaT-BPP for improved performance, as we illustrate for the SOTA case of linear basis (Deng & Russakovsky, 2022) in Section 4.2.

**Setup** Building upon existing literature, we employ standard ConvNet architectures (Zhao & Bilen, 2021b; Deng & Russakovsky, 2022; Cazenavette et al., 2022) —three layers for $32 \times 32$ images and four layers for $64 \times 64$ images. Our distilled data is trained utilizing Algorithm 1, with the Higher package (Grefenstette et al., 2019) aiding in the efficient calculation of meta-gradients. We opt for a simple setup: using Adam for inner optimization with a learning rate of 0.001, and applying standard augmentations (flip and rotation) on the target set. Parameters such as unrolling length and window size are determined via a validation set.

**Evaluation** During the evaluation phase, we adhere to the standard augmentation protocol as per Deng & Russakovsky (2022); Zhao & Bilen (2021b). We conduct evaluations of each distilled data set using ten randomly selected neural networks, reporting both the mean and standard deviation of the results. For all other baseline methodologies, we record the best value reported in the original paper. Note that Zhou et al. (2022); Loo et al. (2023) employs a 4 or 8 times wider ConvNet to reduce discrepancies between surrogate approximations and actual training. To ensure alignment with this protocol, we provide a transfer evaluation of our method, that is we distill with a narrow network and evaluate with a wide network. We also re-evaluate their checkpoints for narrow networks. Complete details can be found in the Appendix. We release our code at `https://github.com/fengyzpku/Simple_Dataset_Distillation`.

Table 1: Performance of different dataset distillation techniques on standard datasets. The **AVG** column denotes the average performance across all the other columns. * denotes works where performance evaluated with wider ConvNets. FRePO and RCIG are the re-evaluated results with narrow networks. Results denotes the best results for narrow networks while results denotes best for wide networks.

| | Dataset | CIFAR-10 | | | CIFAR-100 | | | CUB200 | | T-ImageNet | | ImageNet-1K | |
| | Img/class(IPC) | 1 | 10 | 50 | 1 | 10 | 50 | 1 | 10 | 1 | 10 | 1 | 2 |
|---|---|---|---|---|---|---|---|---|---|---|---|---|---|
| Inner Loop | BPTT (Deng & Russakovsky, 2022) | 49.1±0.6 | 62.4±0.4 | 70.5±0.4 | 21.3±0.6 | 34.7±0.5 | - | - | - | - | - | - | - |
| | KIP* (Nguyen et al., 2021b) | 49.9±0.2 | 62.7±0.3 | 68.6±0.2 | 15.7±0.2 | 28.3±0.1 | - | - | - | - | - | - | - |
| | RFAD* (Loo et al., 2022) | 53.6±1.2 | 66.3±0.5 | 71.1±0.4 | 26.3±1.1 | 33.0±0.3 | - | - | - | - | - | - | - |
| | FRePO* (Zhou et al., 2022) | 46.8±0.7 | 65.5±0.6 | 71.7±0.2 | 28.7±0.1 | 42.5±0.2 | 44.3±0.2 | 12.4±0.2 | 16.8±0.1 | 15.4±0.3 | 25.4±0.2 | 7.5±0.3 | 9.7±0.2 |
| | FRePO | 45.6±0.1 | 63.5±0.1 | 70.7±0.1 | 26.3±0.1 | 41.3±0.1 | 41.5±0.1 | - | - | 16.9±0.1 | 22.4±0.1 | - | - |
| | RCIG* (Loo et al., 2023) | 53.9±1.0 | 69.1±0.4 | 73.5±0.3 | 39.3±0.4 | 44.1±0.4 | 46.7±0.1 | 12.1±0.2 | 15.7±0.3 | 25.6±0.3 | 29.4±0.2 | - | - |
| | RCIG | 49.6±1.2 | 66.8±0.3 | - | 35.5±0.7 | - | - | - | - | 22.4±0.3 | - | - | - |
| Modified Objectives | DSA (Zhao & Bilen, 2021b) | 28.8±0.7 | 52.1±0.5 | 60.6±0.5 | 13.9±0.3 | 32.3±0.3 | 42.8±0.4 | 1.3±0.1 | 4.5±0.3 | 6.6±0.2 | 14.4±2.0 | - | - |
| | DM (Zhao & Bilen, 2023) | 26.0±0.8 | 48.9±0.6 | 63.0±0.4 | 11.4±0.3 | 29.7±0.3 | 43.6±0.4 | 1.6±0.1 | 4.4±0.2 | 3.9±0.2 | 12.9±0.4 | 1.5±0.1 | 1.7±0.1 |
| | MTT (Cazenavette et al., 2022) | 46.3±0.8 | 65.3±0.7 | 71.6±0.2 | 24.3±0.3 | 40.1±0.4 | 47.7±0.3 | 2.2±0.1 | - | 8.8±0.3 | 23.2±0.2 | - | - |
| | TESLA(Cui et al., 2023) | 48.5±0.8 | 66.4±0.8 | 72.6±0.7 | 24.8±0.4 | 41.7±0.3 | 47.9±0.3 | - | - | - | - | 7.7±0.2 | 10.5±0.3 |
| | FTD (Du et al., 2023) | 46.8±0.3 | 66.6±0.3 | 73.8±0.2 | 25.2±0.2 | 43.4±0.3 | 50.7±0.3 | - | - | 10.4±0.3 | 24.5±0.2 | - | - |
| | Ours | 53.2±0.7 | 69.4±0.4 | 75.3±0.3 | 36.3±0.4 | 47.5±0.2 | 50.6±0.2 | 13.8±0.3 | 17.7±0.2 | 22.1±0.3 | 24.4±0.2 | 10.77±0.4 | 12.05±0.5 |
| | Ours (transfer to wide) | 54.1±0.4 | 71.0±0.2 | 75.4±0.2 | 36.9±0.3 | 47.9±0.2 | 51.0±0.3 | 14.2±0.3 | 17.9±0.4 | 22.5±0.1 | 24.9±0.1 | 11.04±0.5 | 12.51±0.5 |
| | Full Dataset | | 83.5±0.2 | | | 55.3±0.3 | | 20.1±0.3 | | 37.6±0.5 | | 33.8±0.3 | |

## 4.1 BENCHMARK PERFORMANCE

Our simple approach to dataset distillation demonstrates competitive performance across a number of datasets (Table 1). With 10 and 50 images per class, our approach gets the state-of-the-art (SOTA) results on the CIFAR-100, CIFAR-10, CUB200, and ImageNet-1K datasets (Table 1 last two rows). Moreover, we achieve these results without any approximations to the inner loop. When considering all IPC values in $\{1, 10, 50\}$, across all datasets, our approach performs as well as the RCIG method up to statistical significance. Encouragingly, even though our bilevel optimization is not biased towards wider networks, we obtain state-of-the-art performance even for wide networks on CIFAR10, CIFAR100, CUB200, and ImageNet-1K across all IPC values. Moreover, when the datasets from the wider-network approaches are evaluated on practical, narrower settings we find that they show a significant drop in performance, going from 39.3% to 35.5% (RCIG, CIFAR100, IPC1) or from 25.4% to 22.4% (FrePO, TinyImageNet, IPC10). Thus, our work generalizes gracefully to wider networks that are used by previous work (improving in performance) as well as narrower networks (for which we can tune directly). This is a significant advantage of our work over prior state-of-the-art.

## 4.2 COMBINATION WITH PARAMETRIZATION METHODS

A separate and complimentary line of work aims to improve the optimization via parameterization of the distilled dataset. Liu et al. (2022); Wang et al. (2023) leverage encoder-decoders, Lee et al. (2022a); Cazenavette et al. (2023) use generative priors, Kim et al. (2022); Liu et al. (2023b) propose multi-scale augmentation, and Deng & Russakovsky (2022) designs linear basis for the dataset.

Note that our performance improvements come from a careful study of the bilevel optimization problem. In principle, RaT-BPTT is complimentary to most of these parameterization ideas and can be seamlessly intergrated. For instance, we adopt the linear basis from Deng & Russakovsky (2022) within our framework. We only study the case of CIFAR10, as for all the other benchmarks RaT-BPP gives better performance even without data parametrization. Without any hyper-

Table 2: Combination of RaT-BPTT with linear parameterization leads to further improvement. We only present those settings where parameterization outperforms the standard RaT-BPTT.

| Dataset | | CIFAR-10 | |
|---|---|---|---|
| Img/class(IPC) | | 1 | 10 |
| Para-meteri-zation | IDC (Kim et al., 2022) | $50.0_{\pm 0.4}$ | $67.5_{\pm 0.5}$ |
| | LinBa (Deng & Russakovsky, 2022) | $66.4_{\pm 0.4}$ | $71.2_{\pm 0.4}$ |
| | HaBa (Liu et al., 2022) | $48.3_{\pm 0.8}$ | $69.9_{\pm 0.4}$ |
| | Linear + RaT-BPTT | $68.2_{\pm 0.4}$ | $72.8_{\pm 0.4}$ |

parameter tuning, we can improve their performance by around 1.6%, leading to astonishing numbers of 68.2% for IPC1 and 72.8% for IPC10 (the numbers w/o parameterization are 53.2% and 69.4% respectively). The results are shown in Table 2.

## 4.3 ABLATIONS ON THE RANDOM TRUNCATED WINDOW

In Section 3, we justify the necessity of performing truncation to speed up and stabilize the gradient, and the necessity of changing the truncated window to cover the entire trajectory. We present an ablation study on selecting the truncated window, comparing random uniform truncation with backward and forward moving methods. The latter two involve initial window placement at the trajectory's start or end and shifting it when the loss stagnates.

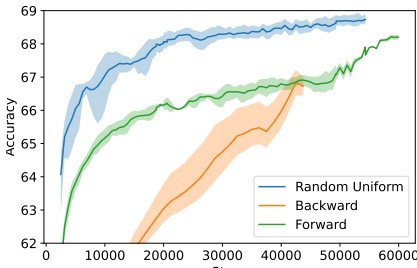

Figure 5: Comparison between random uniform truncation, backward moving, and forward moving. Random uniform truncation gives the best performance across the whole training process. N=120, T=40 for IPC10 with CIFAR10.

Figure 5 reveals that random uniform truncation outperforms other methods throughout training. A closer examination of the forward and backward moving curves suggests that altering the window's positioning can spur noticeable enhancements in accuracy. This suggests that different truncation windows capture unique aspects of knowledge, emphasizing the need for comprehensive trajectory coverage. Actually, uniform sampling isn't the optimal design. By

focusing more on the initial phase, accuracy on CIFAR10 with IPC10 improves by 0.4%, though it requires fine-tuning an extra hyper-parameter. For simplicity, we opt for the uniform approach.

## 5 INTERCORRELATIONS AND BOOSTING

Having established the strength of our method, we proceed to dissect the structure of the learned datasets to catalyze further progress. Nearly all the current distillation method optimize the data jointly. Such joint optimization often lead to unexpected behaviors which are largely absent in the original dataset. For instance, Nguyen et al. (2021b) observes that the intrinsic dimension of distilled data increases, compared to the training data. The data is jointly learned, and therefore can be *correlated*. A particularly noticeable consequence of the correlation is a significant drop in performance when using a subset of the distilled data for downstream tasks, as shown in Figure 6. We see that subsampling 5 images per class from an IPC50 distilled dataset not only gives performance way below an IPC5 dataset learned from scratch, but performs even worse than training on 5 *random* training images per class.

To address these challenges, we propose *boosted* dataset distillation (Boost-DD) (in Algorithm 2). Boost-DD controls inter-correlation within distilled data and can be integrated with any gradient-based dataset distillation algorithm. The central idea is to construct the distilled data iteratively with smaller data groups called "blocks" as illustrated in Figure 2. For IPC50, we divide all images into blocks of IPC5. We start from a distilled IPC5. Each time, we add another fresh block of IPC5, and optimize the new block with reduced learning rate on the existing blocks by a factor of $\beta \leq 1$. The extreme case where the learning rate is zero for previous blocks is termed *strongly-boost* ($\beta = 0$). The advantage of this approach is that initial blocks remain unchanged when the new block is added, ensuring consistent performance. This results in a "nested" dataset where subsampling earlier blocks yields high-performing subsets! We call the algorithm *weakly*-boost $\beta$ is non-zero and perform experiments with $\beta = 0.1$. When performing distillation for each fix IPC setting, we follow the same configurations as our RaT-BPTT.

---

**Algorithm 2** Boosted Dataset Distillation (Boost-DD)

---

**Input:** Target dataset $\mathcal{D}$. Distillation-Algorithm $\mathcal{A}$ with initiatlization procedure $\mathcal{I}(size)$ and meta-learning rate $\alpha$, outputting distilled data $\mathcal{U} = \mathcal{A}(\mathcal{D}, \mathcal{I}(size))$ ($|\mathcal{U}| = size$). Block size $b$. Number of blocks $J$. Boosting-strength $\beta \in [0, 1]$.
**Output:** Distilled data $\mathcal{U}$ with $|\mathcal{U}| = b \cdot J$.

1: Distill first block of size $b$: $\mathcal{U}_0 := \mathcal{A}(\mathcal{D}, \mathcal{I}(b))$.
2: **for** $j = 1 \ldots J - 1$ **do**
3:   Distill increased data $\mathcal{U}_j = \mathcal{A}(\mathcal{D}, \mathcal{U}_{j-1} \cup \mathcal{I}(b))$ using "stale" meta-learning rate $\alpha_s = \beta \cdot \alpha$ on the first $(j-1) \cdot b$ data points and $\alpha$ on the last $b$.
4: **end for**
5: $\mathcal{U} := \mathcal{U}_{J-1}$.

---

Figure 7 shows how weakly-boost and strongly-boost compare to distilling each of the sub-datasets from scratch. It is important to highlight that the curve for 'strongly-boost' represents the accuracy curve obtainable when subsampling across different budgets. We observe that even strongly-boost

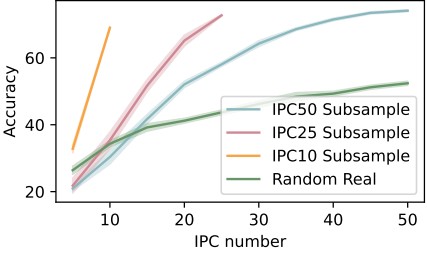

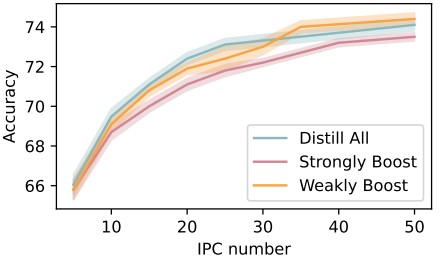

Figure 6: Test performance of random subsamples of larger distilled datasets, compared to performance of real data of the same size. We average over 10 random samples. Setting: CIFAR10, RaT-BPTT.

Figure 7: Performance of fully distilled, strongly and weakly boosted distilled data. Boosting essentially retains performance compared to jointly distilled data. CIFAR10, RaT-BPTT.

results in exceedingly minor sacrifice in performance compared to joint distillation from scratch, especially with larger distilled datasets. In Appendix D.9 we present a visual comparison of images distilled jointly with images distilled with boosting. Boosted images seem to show larger diversity and seem closer to real images. In Appendix D.8, we further combine our framework with other distillation methods like the MTT.

# 6 DISCUSSION

In this work, we proposed a simple yet effective method for dataset distillation, based on random truncated backpropagation through time. Through a careful analysis of BPTT, we show that randomizing the window allows to cover long dependencies in the inner problem while truncation addressed the unstable gradient and the computational burden. Our method achieves state of the art performance across multiple standard benchmarks, across both narrow as well as wide networks. We note that the utility of the RaT-BPTT approach has the potential to extend to a wider variety of bilevel optimization problems, for instance to meta-learning of hyperparameters (Andrychowicz et al., 2016), MAML-like problems (Finn et al., 2017a) for good few-shot adaptation to new tasks, and Poisoning Attacks (Muñoz González et al., 2017) for crafting training-data to corrupt a system's test behavior. All these problems have a bilevel optimization at its core which, in general, requires optimization through unrolling and is thus amenable to the RaT-BPP approach.

Nonetheless, several design choices are guided by intuitions and observations, leaving room for improvement. For instance, we could envision adopting methods from Maclaurin et al. (2015) to improve efficiency of meta-gradient computation or from Sachdeva et al. (2023) for algorithmic design for Adam with constant memory consumption. Integrating such methods into our framework presents an exciting avenue for developing a scalable algorithm. This integration could potentially enhance the computational efficiency and memory management of our method, an aspect we are keen to explore in future work. We defer a detailed limitation discussion to Appendix A.

Further, we address the catastrophic degradation in performance of subsets of distilled data with our boosting method. It allows us to create a single versatile dataset for various distillation budgets with minimal performance degradation. However, the boosted dataset still has inner correlation between blocks. This is evident in Figure 8 when comparing the performance of the first IPC5 block with the second one obtained via strongly-boost (though both of them are much higher than sampling a random IPC5 from the jointly distilled IPC50). Moreover, as shown in Figure 7, weakly-boost for larger IPC eventually outperforms joint training. Since weakly-boost generates less inter-correlated datasets, this hints at the possibility that strong intercorrelations are one reason for diminishing returns observed when increasing the size of distilled datasets. While higher-order correlations may potentially encode more information, they also compromise data interpretability, diverging from the standard IID paradigm. Is it possible to further minimize these correlations, especially in larger IPC datasets? We leave these questions to future research.

We also attempt to understand other factors bottlenecking the gains when scaling up distilled data. Specifically, we try to understand what information is learnt in the distilled data by dissecting the accuracy on evaluation samples. We leverage a hardness score that characterizes whether data is easy or hard, stratify the accuracy by hardness and compare it for the original network and the network trained on distilled data for a range of IPC in Figure 9 (details in Appendix E). One would have hoped that adding more distilled data would help to distill more of the hardness tail, but this is not the case. This suggests that future work might benefit from focusing on how one can distill data that is better adapted to larger hardness scores, for instance by infusing validation batches with harder data, placing emphasis on the middle of the distribution. A preliminary promising study is presented in Appendix E.

## 7 ACKNOWLEDGEMENTS

This work was supported by the National Science Foundation under NSF Award 1922658. This work was also supported in part through the NYU IT High Performance Computing resources, services, and staff expertise. The authors extend their gratitude to all reviewers for their insightful suggestions. YF would like to thank Di He, Weicheng Zhu, Kangning Liu, and Boyang Yu for discussions and suggestions.

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

## A    LIMITATIONS

**Algorithm Design** The design of our method is primarily guided by intuitions and observations from empirical studies. Throughout the algorithm's development, we aim to strike a balance between scalability and effectiveness. Our approach currently involves tuning the ratio between the unrolling length and window size, scaling the unrolling length in accordance with the IPC number and GPU size. While this approach has demonstrated promise, we acknowledge that the current algorithmic choice might not represent the absolute optimal solution. Further research could investigate alternative algorithm designs.

**Application to larger models and datasets** A notable strength of our methodology is its versatility: it is compatible with all differentiable loss functions and network architectures, emphasizing its broad applicability. However, we only focus on illustrating the method's capabilities with standard benchmarks in the literature. This decision leaves a promising avenue for future work to apply and validate our method across various domains and tasks beyond image classification. It's also worth highlighting that while surrogate-based techniques are constrained to using the MSE loss to convexify the inner problem, our approach is agnostic to the specific loss function employed. This flexibility paves the way for our method's application in other realms, such as audio and text data distillation.

**GPU memory usage** Despite the significant improvements introduced by RaT-BPTT, it still necessitates unrolling and backpropagating over several steps, which require storing all intermediate parameters in the GPU. Consequently, this method incurs substantial memory consumption, often exceeding that of directly training the model. For larger models, one might need to implement checkpointing techniques to manage memory usage effectively.

## B    FIGURES FOR DISCUSSION

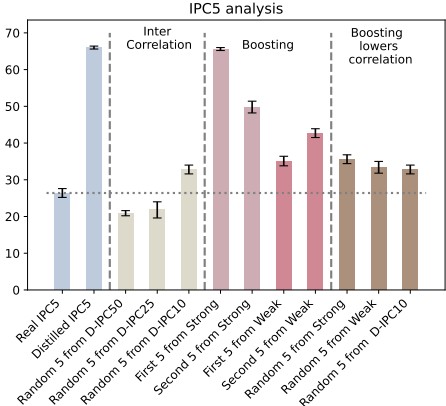

Figure 8: Performance of subsamples compared to fully distilled and random real images for IPC5: y-axis shows test accuracy a) Random IPC5 from IPC50 and IPC25 performs worse than random *real* IPC5, indicating the strong inter-correlation learned in the dataset. b) IPC5 building blocks of boosting perform quite well. c) Random IPC5 from boosted IPC10 performs better than random IPC5 from standard IPC10. Boosting somewhat lowers the inter-correlation.

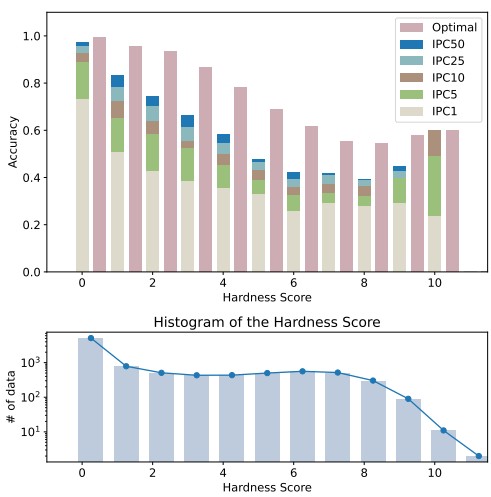

Figure 9: **Top:** Hardness score vs accuracy of dataset distillation (for various images per class (IPC)). "Optimal" indicates accuracy when training on the entire real data. **Bottom:** Histogram of the hardness scores. Notice how score 0 (unforgettable) examples are distilled well (top), but harder examples are progressively harder to distill. Details in Appendix E

## C    OTHER RELATED WORK

In this section, we discuss further works related to dataset distillation or hardness metrics.

*Boosting*: It is noteworthy that Liu et al. (2023a) has also identified challenges associated with retraining distilled datasets for varying budgets. Their proposed solution adopts a top-down approach, aiming to slim a large distilled dataset. In contrast, our method follows a bottom-up strategy, producing a modular dataset designed to accommodate various budgets. Moreover, one of our motivations is to address and study the intercorrelation problem.

*Hardness metrics:* One way to study generalization performance of neural nets is to understand which data points are "easy" or "hard" to learn for a neural network. There is an intimate relationship to *data pruning* which tries to understand and quantify which subsets of the data can be pruned with impunity, while maintaining the performance of the neural net when trained on the remainder[1]. Inspired by the phenomenon of catastrophic forgetting, Toneva et al. (2019) are the first to study how the learning process of different examples in the dataset varies. In particular, the authors analyze whether some examples are harder to learn than others (examples that are forgotten and relearned multiple times through learning.) and define a *forgetting score* to quantify this process. To our knowledge, our work is the first to use this tool to understand how learning on distilled data differs from learning on full data, to identify bottlenecks. The idea to "enrich" the validation data during the data distillation process appears in Liu et al. (2023b), who chose more "representative" data to learn from, as determined by k-means clustering. To our knowledge, we are the first to propose learning from "harder-to-learn" data towards more efficient data distillation.

*Extensions of Dataset Distillation:* Beyond the conventional dataset distillation formula that aims to minimize generalization error, there have been advances in optimizing metrics for various objectives. These include dataset generation tailored for generalization attacks Yuan & Wu (2021), adversarial perturbations Tsilivis & Kempe (2022), and generating distilled data with an emphasis on robustness Tsilivis et al. (2022).

# D  EXPERIMENTS

## D.1  EXPERIMENTAL DETAILS

**Data Preprocessing**: Leveraging a regularized ZCA transformation with a regularization strength of $\lambda = 0.1$ across all datasets except only the ImageNet-1K, our approach adheres to the methods established by prior studies Nguyen et al. (2021a;b); Zhou et al. (2022); Loo et al. (2022); Deng & Russakovsky (2022). We apply the inverse ZCA transformation matrix for distillation visualization, using the mean and standard deviation to reverse-normalize optimized data.

**Models** Following previous works, we use Instance Normalization for all networks for both training and evaluation.

**Initialization** In contrast to conventional real initialization widely used in nearly all previous works, we employ random initialization for distilled data, hypothesizing that there is a reduction in bias from such uninformative initialization. Data are initialized via a Gaussian distribution and normalized to norm 1. For RaT-BPTT, we note comparable performance and convergence between random and real initialization.

**Label Learning** Following previous works that leverage learnable labels, we optimize both the data and label for CIFAR10 IPC10 and IPC50, all IPCs for CIFAR100, CUB-200, Tiny-ImageNet, and ImageNet-1K. We forego normalization for label probability, hence the labels retain their positive real value representation.

**Training** In addition to the RaT-BPTT algorithm, we incorporate meta-gradient clipping with an exponential moving average to counter gradient explosion. We find that the proper combination of normalizing initialization and learning rate (0.001 for Adam) is crucial for successful distillation image training. While using instance normalization, an image scaled by $\alpha$ leads to meta-gradient scaling by $\frac{1}{\alpha}$. As a result, one should use an $\alpha$ times larger learning rate for Adam or $\alpha^2$ times larger for SGD to achieve the same optimization trajectory. We thus adopt a similar initialization scale to that of neural network training (normalized to norm 1), combined with a standard learning

---

[1]The boundary between *coresets* and *data pruning* is fluid; the former term is used for small subsets of the training set, while the latter usually refers to removing only a fraction of the training data, like $25\%$ Paul et al. (2021).

rate of 0.001 when using Adam. To maintain meta-gradient stability, we employ batch sizes of 5,000 for CIFAR-10 and CIFAR-100, 3,000 for CUB-200, 1,000 for Tiny-ImageNet, and 1,500 for ImageNet-1K. Note that one should aim to further increase the batch size for Tiny-ImageNet until all the GPU memory is used. For sampling from the distilled data, a batch size of 100 or 200 samples is adopted. This translates to 10 samples per class in CIFAR10 (10 classes), 1 sample per class in CIFAR100 (100 classes, except for IPC 50 where it's 2 samples per class), and 1 sample per class in CUB200 (200 classes). For TinyImageNet and ImageNet-1K, due to GPU limitations, we use 1 sample per class from 50 randomly selected classes in each batch. Ideally, one should further increase the number of classes for these large scale dataset.

**Hyperparameters** In an effort to minimize tuning requirements, we adhere to a standard baseline across all configurations. Specifically, we utilize the Adam optimizer for both the inner loop (network unrolling) and the outer loop (distilled dataset optimization) with learning rates uniformly set to 0.001 for CIFAR-10, CIFAR-100, and CUB-200, and to 0.0003 for Tiny-ImageNet and ImageNet-1K. We refrain from applying weight decay or learning rate schedules that are used in prior works Zhou et al. (2022); Loo et al. (2023).

**Evaluation** We evaluate our optimized data using a seperate held-out test dataset (the test set in the corresponding dataset). We adopt the same data augmentation as in previous work Deng & Russakovsky (2022). We train standard convolutional networks using the Adam optimizer with a learning rate of 0.001. For wide convolutional networks, we employ the same optimizer but adjust the learning rate to 0.0003. No learning rate schedule is used. During the evaluation phase, we maintain the same batch size for the distilled data as used in the training phase.

## D.2 ABLATIONS ON CURRICULUM

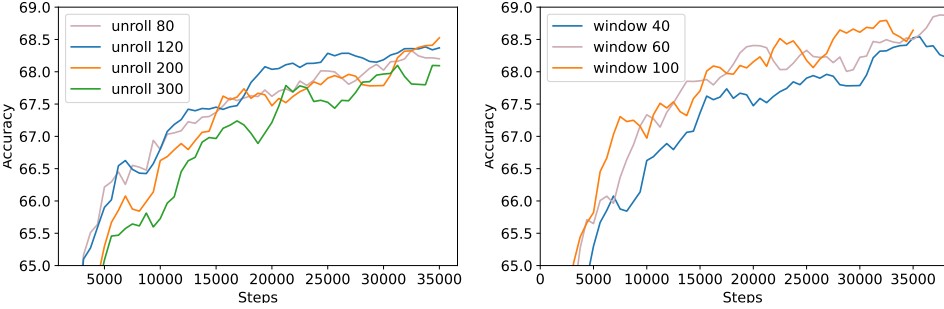

Figure 10: **Left** Test accuracy during distillation for different unrolling length of 80, 120, 200, 300 with fixed window size 40. CIFAR-10, IPC10. **Right** Test accuracy during distillation for different window size in 40, 60, 100 with fixed unrolling length 200.

Our RaT-BPTT implementation hinges on tuning two hyperparameters: unrolling length and back-propagation window size. This section presents an ablation study exploring these parameters for CIFAR-10, IPC10.

**Unrolling length**

We initially fix the window size at 40 while varying the unrolling length. Notably, unrolling length governs the long-term dependencies we can capture within the inner loop. Figure 10 reveals that a moderate unrolling length, between twice and five times the window size, yields similar performance. However, disproportionate unrolling, as seen with a window size of 40 and unrolling length of 300, detrimentally affects performance.

**Window size**

Next, we fix the unrolling length at 200 and experiment with window sizes of 40, 60, and 100. Figure 10 shows the latter two sizes yield comparable performance. In RaT-BPTT, GPU memory consumption is directly proportional to the window size, thus a window size of 60, with an unrolling length of 200 steps, emerges as an optimal balance. As such, we typically maintain a window size to unrolling length ratio of around 1:3.

In our implementation, we employ a window size and unrolling length of (60, 200) for CIFAR-10 IPC1 and CUB-200, (80, 250) for CIFAR-10 IPC10, and (100, 300) for all other datasets.

### D.3 OTHER METRICS ON GRADIENT STABILITY

In Figure 3, we investigated the stability of meta-gradients using gradient norms as a metric, predicated on the notion that stable and efficient learning should manifest as consistent and decreasing gradient norms throughout training. Expanding on this analysis, we now introduce another metric for evaluating gradient stability: the normalized gradient variance, in line with the methodology proposed by Faghri et al. (2020). Each variance value reflects the instability across the batch samples, and the values across time steps reflects the instability across training steps.

To calculate this metric, we compute the average variance of all gradient entries using a set of 100 samples from the evaluation batch. Given the different scales in gradient norms across different methods, we normalize this variance against the square of the norm. This normalization yields a more consistent metric, termed the normalized variance. Employing the same experimental setup as in Figure 3, we present the results in Figure 11. It shows that RaT-BPTT not only maintains lower variance at each training step but also demonstrates more consistent variance trajectories over the course of training. These findings, in conjunction with the earlier results from Figure 3, collectively offer a comprehensive view of the argued training instability.

### D.4 ABLATION ON STABILIZING BPTT

In Figure 3, we have demonstrated the notable instability of meta-gradients via the gradient norm. This section extends our analysis with ablation studies, indicating that both controlling the gradient norm and incrementally increasing the unrolling parameter $T$ of BPTT result in only marginal improvements, which cannot compare to the gains garnered through RaT-BPP. We follow the setting in Figure 4.

Our foundational approach has already incorporated gradient clipping to manage extreme gradient norm values, employing a standard exponential moving average (EMA) with a $0.9$ decay rate and capping the gradient norm at twice the adaptive norm.

To further stabilize the gradient norm, we explored two additional methods: 1) BPTT-Gradient Clipping, limiting the gradient norm to no more than 1.05 times the adaptive norm, and 2) BPTT-Normalized Gradient, ensuring a consistent gradient norm of 1 throughout training. However, as Figure 12 illustrates, these methods achieve only marginal enhancements over the basic BPTT approach. Their performance trails behind RaT-BPTT, with a threefold increase in optimization time due to extended backpropagation.

These findings highlight challenges such as deviation from the kernel regime, variance from extensive unrolling, and pronounced non-convexity, contributing to gradient instability, as evidenced by fluctuating gradient norms. Addressing these issues solely by adjusting gradient norms proves insufficient.

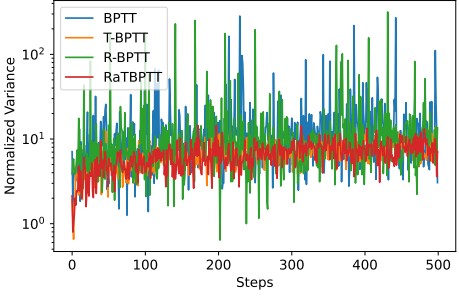
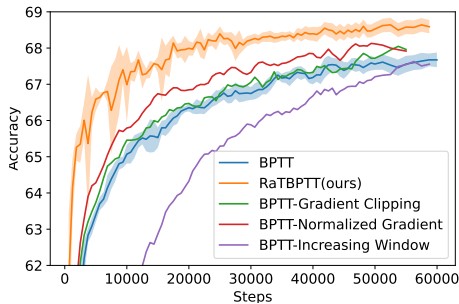

Figure 11: Normalized variance across batch samples of the meta-gradient. RaT-BPTT has stable and small variances. Same setting as in Figure 3.

Figure 12: Ablations on stabilizing BPTT: only controlling the gradient norm or gradually increasing the window is not enough.

Alternatively, we examine limiting the maximum Hessian matrices in Equation (2) by gradually extending the unrolling length $T$ in BPTT. In Figure 12, the BPTT-Increasing Windows variant, which linearly scales $T$ from 10 to 180, underperforms both R-BPTT and standard BPTT. This underlines the complexity within the inner loop, deviating significantly from the kernel regime and emphasizing the importance of managing the unrolling window size.

## D.5 OTHER ARCHITECTURES

Table 3: Generalization to other architectures. We conduct transfer evaluation of a distilled dataset trained with a 3-layer ConvNet and directly training the dataset with the architecture in the inner loop. CIFAR10, IPC10.

| Architecture | VGG-11 | AlexNet | ResNet-18 |
|---|---|---|---|
| Transfer | 46.6±0.9 | 60.1±0.6 | 49.2±0.8 |
| Direct Training | 47.7±0.8 | 63.7±0.6 | 53.0±0.8 |

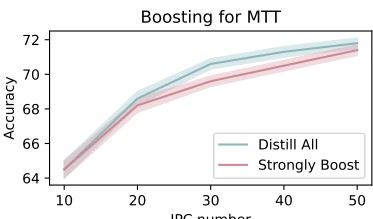

Figure 13: Boosting with MTT

We further assessed our method across various architectures to demonstrate its universality. It is noteworthy that our approach is already effective across different widths of the convolutional networks (narrow and wide) we used. Additionally, we conducted tests using the standard VGG-11, AlexNet, and ResNet-18, both training it from scratch and transferring from the distilled dataset. To our knowledge, we are the pioneers in applying direct distillation to a standard-sized network like ResNet-18 and VGG-11. Prior works never train directly on VGG11 and they only use small or modified ResNets like ResNet-10 (Zhang et al., 2022a; Kim et al., 2022), ResNet-12 (Deng & Russakovsky, 2022) and ResNet-AP10 (Kim et al., 2022; Liu et al., 2023b) in these settings.

The results are presented in Table 3. Our results yield better or comparable transfer results compared with previous methods. Direct training further increases the numbers.

## D.6 ABLATION ON THE INNER OPTIMIZER

We have opted for Adam instead of SGD to simplify the tuning process for the inner loop. This decision was based on the ability to use a common learning rate without requiring decay in the inner loop. In this section, we perform ablation studies on how the inner loop optimizer affects the performance.

We implement RaT-BPTT (SGD) using SGD with learning rate 0.01 and learning rate decays at [120, 200] by 0.2. For IPC10 on CIFAR10, RaT-BPTT (SGD) achieves a 69.0% accuracy (std 0.3%), while RaT-BPTT (Adam) results in a slightly higher accuracy of 69.4% (std 0.4%). Thus, RaT-BPTT (SGD) also outperforms previous methods in this setting by a large margin. It is crucial to note that our improvement is attributed to factors beyond merely employing Adam in the inner loop.

It is also noteworthy to point out that we are not the first to use Adam for the inner loop during training. Loo et al. (2022) also uses Adam for their linearized inner loop. Some other papers (Loo et al., 2023; Zhou et al., 2022) have also adopted Adam for the linear loop during evaluations. We suspect that whenever Adam was an option, the benchmarking papers probably tried it without significant improvements.

## D.7 DISCUSSIONS ON EFFICIENCY

We have conducted a comparative analysis of the total training time for several methods, utilizing a consistent computational environment on an RTX8000 with 48GB. It should be noted that we have excluded RCIG from this comparison, as our reproduced accuracy falls short of the reported number. The following are the recorded training times (in hours) for CIFAR10 with IPC10: KIP (over 150), LinBa (100), FrePO (8), RaT-BPTT (22), MTT (14), and IDC (35). Among these methods, our cost ranks as the third best.

There are ways to further improve the efficiency. 1) The current package we utilize for meta-gradient calculation, the higher package, as noted in Maclaurin et al. (2015), lacks efficiency compared to other methods. We could lower the time cost by altering our implementation to more efficient methodologies. 2) The references (Maclaurin et al., 2015; Sachdeva et al., 2023) contain efficient designs for the meta-gradient calculation. As reported in Sachdeva et al. (2023), it could lead to up to 2x speedups compared with the higher package. This improvement would not only enhance the performance of our method but also bring it in line with the efficiency benchmarks set by methodologies like FrePO. 3) Similar to FrePO, we may keep a pool of parameter checkpoints to further optimize our method. This strategy would reduce the need for inner training from new random initializations.

## D.8    BOOSTING FOR OTHER METHODS

Our boosting framework is a method that can be combined with any gradient-based dataset distillation method. In this section, we combine boosting with MTT Cazenavette et al. (2022) as a proof of principle, choosing a representative method. We implement strong boosting (with $\beta = 0$) for MTT from IPC 10 to IPC 50 in steps of IPC 10. In Figure 13, the final generalization accuracy is 71.4%. The accuracy improves rapidly throughout the boosting and the final performance is marginally lower (exact percentage pending) than the direct distillation of MTT, which stands at 71.6%.

## D.9    VISUALIZATION

We incorporate visualizations for IPC10 on CIFAR-10, representing standardly trained (Figure 14), weakly boosted (Figure 16), and strongly boosted images (Figure 15). Upon inspection, the images from both boosted categories appear more diverse compared to their standard counterparts.

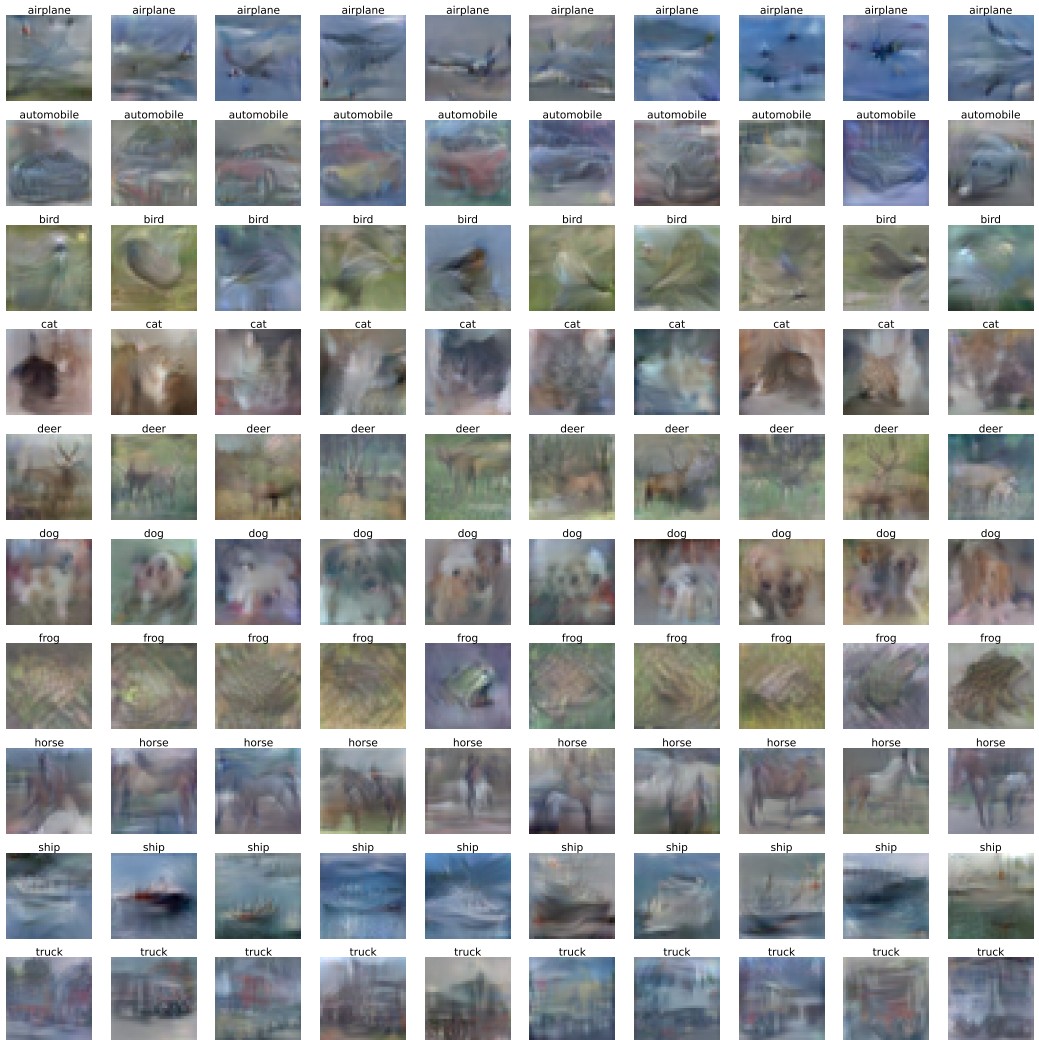

Figure 14: Visualization for RaT-BPTT standardly trained on CIFAR-10 with IPC10.

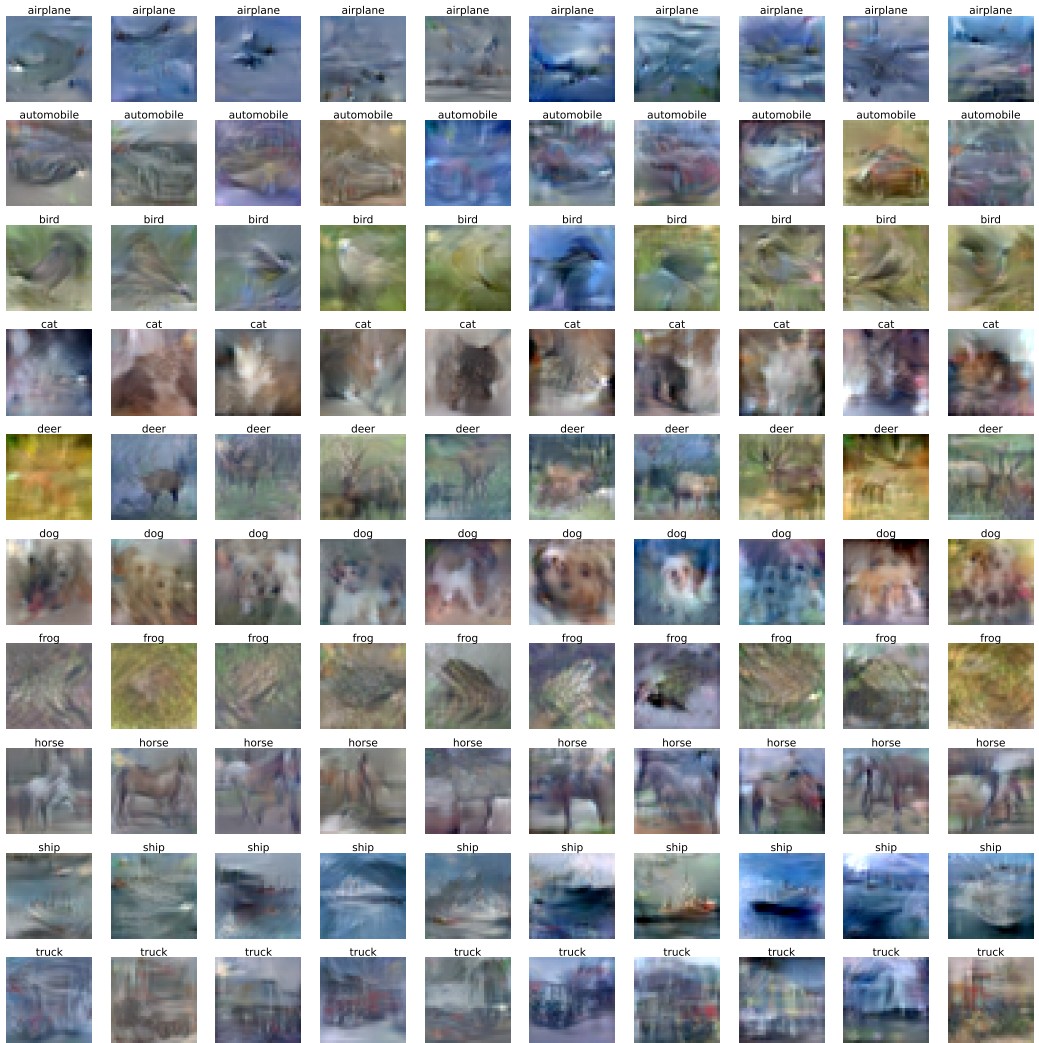

Figure 15: Visualization for RaT-BPTT with strong boosting (Boost-DD). CIFAR-10 with IPC10.

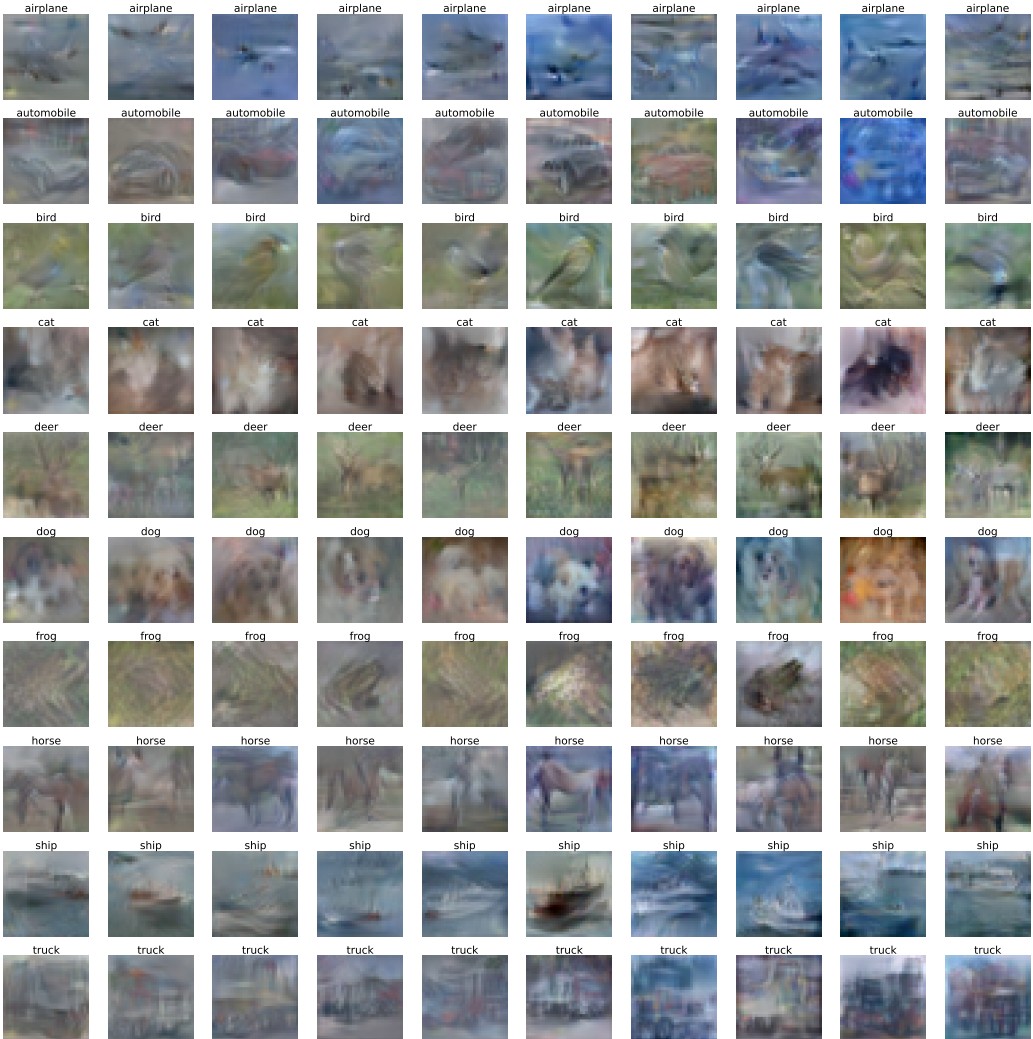

Figure 16: Visualization for RaT-BPTT with weak boosting (Boost-DD). CIFAR-10 with IPC10.

# E    HARDNESS ANALYSIS

## E.1    DISSECTING THE DATA

Now, we attempt to understand what is bottlenecking the gains when scaling up the distilled data from say IPC1 to IPC50 (see Table 1) by analyzing the performance of dataset distillation on examples that are easy or hard to learn, using a *hardness score* to stratify the data. We first leverage the *forgetting score* Toneva et al. (2019) as an empirical implementation of the hardness score as it characterizes the difficulty of learning a datapoint along the *training* trajectory.

**Forgetting score.** For each data point $x$ and a network trained for $T$ epochs, we say $x$ has a *forgetting event* at time $t$ if $x$ is correctly classified at time $t$ and misclassified at time $t + 1$. The *forgetting score* of a data point is the sum of its forgetting events before time $T$, averaged over 10 random initializations. Toneva et al. (2019) show that based on the forgetting score a significant amount of training data can be omitted, without sacrificing test performance.

We first compute the forgetting score of the original network for each training data point. We then stratify the accuracy by forgetting score and compare it for the original network and the network trained on distilled data (Figure 9). Notice that the highest boost in performance, especially for the easy datapoints (score 0) comes from simply having one image per class (IPC1). Further, one would have hoped that increasing the number of images per class would help distill more of the tail, enabling models to generalize better to data points with larger forgetting scores. We notice that this happens to some extent till a score of 4, but after that despite there being a lot of datapoints with a larger hardness score (Figure 9, bottom) IPC50 seems to yield minimal marginal improvements. This suggests that future works might benefit from focusing on how one can distill datapoints with larger forgetting scores.

## E.2    HARDNESS SAMPLER

We present an initial approach to enhance learning on challenging examples through a "hardness sampler" that modifies the data batch distribution. Specifically, our objective is to enrich validation batches with more challenging examples, concentrating primarily on those lying mid-way between the extremes of very easy and very hard examples. This approach is inspired by the parabolic shape depicted in Figure 9.

However, the calculation of the forgetting score is often computationally demanding and thus may not be practical for all applications, especially as part of the outer loop in dataset distillation. Moreover, Forgetting score has only been defined for and analysed on networks that are trained on the data that is being scored. To address these challenges, we propose an **adaptive hardness score** that is both efficient and versatile. This score is computed based on the disagreement in predictions Feng et al. (2020) across 8 randomly trained networks using the current distilled dataset. To stay relevant to the evolving challenges, this score is updated adaptively every 50 epochs.

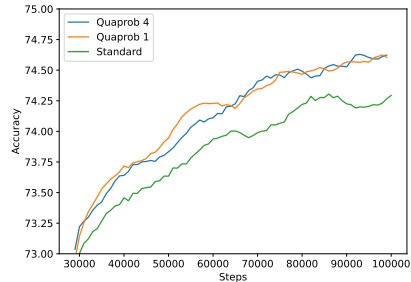

Figure 17: Test performance of hardness sampler versus standard for thresholds 1 and 4. We start to sample at 25K steps since we believe it becomes more relevant in the later stages of distillation. Setting: CIFAR10, IPC50, RaT-BPTT.

Based on this adaptive hardness score, we down-weight the easiest and hardest examples by giving the following weight $w$ to a sample $x$ with hardness score $HS(x) \in \{0, \ldots, 8\}$:

$$w(x) = thr + abs(HS(x) - 4),$$

where $thr$ is a threshold which we set to either $1$ or $4$. For each update of the meta-gradient we sample from a training data point $x$ proportional to $w(x)$, which upweights medium-hard examples the most. Figure 17 demonstrates a notable performance improvement for the IPC50 distillation on CIFAR10, and hints that this direction might be fruitful for future work to pursue.

