# OpenReview forum: "Embarrassingly Simple Dataset Distillation"
_ICLR.cc/2024/Conference — ICLR 2024 poster_

### Official Review · Reviewer_wwkm · 2023-10-30

**Soundness:** 3 good
**Presentation:** 3 good
**Contribution:** 3 good
**Rating:** 6
**Confidence:** 3

**Summary:**

This paper proposes simple and effective method for dataset distillation. Since Back Propagating Through Time (BPTT) is computationally expensive, we truncate a trajectory of inner  optimization steps, called truncated BPTT. However, the performance of truncated BPTT is worse than BPTT due to the biased gradient. In order to get better trade-off between computational cost and performance, the paper proposes to sample random window of inner trajectory, which uses the same amount of computational cost as the truncated BPTT but with better performance. Moreover, it proposes "Boosted Dataset Distillation" to mitigate intercorrelation between distilled images, where it iteratively learns to distill a dataset into a small set while setting the learning rate of previously distilled instances to small value.

**Strengths:**

- The proposed method is simple and effective. It outperforms most of the baselines on CIFAR, CUB and Tiny-ImageNet datasets.

- Ablation studies show the effective of proposed method.

- The proposed method can be easily integrated with other bilevel optimization based dataset distillation methods.

**Weaknesses:**

- Although the proposed method uses the same amount of computational cost as the truncated BPTT method, it still requires to store a trajectory of length $M$. If the dataset becomes large, we may need large $M$ for convergence at inner loop, which hinders scalability of the proposed algorithm.

- It is hard to analyze why the proposed method helps improving performance compared to BPTT or truncated BPTT.

- The paper requires more comprehensive empirical experiments to verify the effectiveness of the proposed method. First and foremost, the authors of the paper did not conduct experiments for architecture generalization. While Table 1 simply demonstrates the method's ability to generalize from a shallow convnet to a wide convnet, I believe this is not adequate. I am curious about how the proposed dataset distillation method can be applied to other architectures, such as VGG, EfficientNet, and ResNet. Secondly, I wonder the proposed method scales to ImageNet dataset which has 1000 classes.

- There is no comparison of computational cost between the proposed method and the other baselines. I think the proposed method is way more expensive than other baselines (such as FrePo) which simplify the  inner loop.

**Questions:**

- What happens if we scale the meta-gradient of BPTT with the norm of the gradient during training? The authors argue training instability of BPTT based on the larger gradient norm. Then the most straightforward way to mitigate the issue would be gradient clipping or scaling the gradient.

- What happens if we gradually increase the total number of unrolling steps for BPTT instead of using fixed $T$?

---

> ### Author Response · Authors · 2023-11-19
>
> Thank you for your valuable suggestions, in particular to clarify computational aspects of our work. This has promted us to conduct an extensive series of additional experiments, as outlined below. We hope that the additional information below speaks for the merits of RaT-BPTT (and might help to determine whether you could raise your score).
> 1. > Although the proposed method uses the same amount of computational cost as the truncated BPTT method, it still requires to store a trajectory of length $M$. If the dataset becomes large, we may need large $M$ for convergence at inner loop, which hinders scalability of the proposed algorithm.
>
> In our experiments, both CIFAR10 with IPC50 (total 500 images) and Tiny-ImageNet with IPC10 (total 2000 images) are tested using the same value of $M$. A moderate $M$ is already large enough for yielding good performance and we observe that the benefits from increasing $M$ significantly diminish after $M=50$. This is evidenced in Figure 10 in the appendix, where we compare the effects of varying  $M$ for CIFAR10 IPC10. The figure demonstrates that increasing $M$ from 40 to 100 only yields small improvements of around 0.6%.
>
> Additionally, we truly appreciate Reviewer dNLL's reference [1] for efficient meta-gradient calculation and reference [2] for algorithmic design for Adam with constant memory consumption. Integrating such methods into our framework presents an exciting avenue for developing a scalable algorithm. This integration could potentially enhance the computational efficiency and memory management of our method, an aspect we are keen to explore in future work.
>
> 2. > It is hard to analyze why the proposed method helps improving performance compared to BPTT or truncated BPTT.
>
> Sorry for the confusion. We have corrected Figure 3 as in the General Response 1. Please also see our discussion on theoretical groundings for RaT-BPTT with Reviewer 1 (Qtcx).
>
>
> 3. > First and foremost, the authors of the paper did not conduct experiments for architecture generalization. While Table 1 simply demonstrates the method's ability to generalize from a shallow convnet to a wide convnet, I believe this is not adequate. I am curious about how the proposed dataset distillation method can be applied to other architectures, such as VGG, EfficientNet, and ResNet.
>
> In our original Appendix C.3 (now C.5), we already include direct training and transfer results for ResNet-18. We further assessed our method across various architectures to demonstrate its universality. As suggested by the reviewer, we conduct experiments for VGG-11, AlexNet, and ResNet-18. The EfficientNet is designed for ImageNet dataset with input size 224 $\times$ 224, and is only applied for finetuning on rescaled CIFAR datasets. Therefore, we choose to incorporate AlexNet instead.
>
> For all these architectures, we conduct training it from scratch and transferring from the distilled dataset with 3-layer ConvNet. To our knowledge, we are the pioneers in applying direct distillation to a standard-sized network like ResNet-18 and VGG-11. Prior works never train directly on VGG11 and they only use small / modified ResNets like ResNet-10 [3,5], ResNet-12 [4] and ResNet-AP10 [5,6] in these settings. We hope these direct training results demonstrate the generality of our method.
>
> | Architecture | VGG-11 | AlexNet | ResNet-18 |
> |----------|----------|----------|----------|
> | Transfer from ConvNet | 46.6\% | 60.1\% | 49.2\% |
> | Direct Training | 47.7\% | 63.7\% | 53.0\% |
>
> Both transfer and direct training achieve good performance.
> We have added these results as Appendix C.5.
>
>
> 4. > Secondly, I wonder the proposed method scales to ImageNet dataset which has 1000 classes.
>
> Most existing works in the dataset distillation field do not directly train on the full ImageNet dataset. However, there are notable exceptions, with two recent works [7, 8] scaling dataset distillation to ImageNet 1K. Given our method's strong performance on the CUB dataset and Tiny-ImageNet, both encompassing 200 classes, we are confident in its scalability to the 1,000 classes in ImageNet-1K. To substantiate this claim, we are happy to include preliminary results on ImageNet 1K in the final camera-ready version of our paper, demonstrating the potential of our approach in handling larger, more complex datasets.

---

> > ### Author Response · Authors · 2023-11-19
> > **Continue**
> >
> > 5. > There is no comparison of computational cost between the proposed method and the other baselines. I think the proposed method is way more expensive than other baselines (such as FrePo) which simplify the inner loop.
> >
> > We have conducted a comparative analysis of the total training time for several methods, utilizing a consistent computational environment on an RTX8000 with 48GB. It should be noted that we have excluded RCIG from this comparison, as our reproduced accuracy falls short of the reported number. The following are the recorded training times (in hours) for CIFAR10 with IPC10: KIP (over 150), LinBa (100), FrePO (8), RaT-BPTT (22), MTT (14), and IDC (35). Among these methods, our cost ranks as the third best.
> >
> > There are ways to further improve the efficiency. 1) The current package we utilize for meta-gradient calculation, the higher package, as noted in [2], lacks efficiency compared to other methods. We could lower the time cost by altering our implementation to more efficient methodologies. 2) The references [1, 2] contain efficient designs for the meta-gradient calculation.  As reported in [2], it could lead to up to 2x speedups compared with the higher package. This improvement would not only enhance the performance of our method but also bring it in line with the efficiency benchmarks set by methodologies like FrePO. 3) Similar to FrePO, we may keep a pool of parameter checkpoints to further optimize our method. This strategy would reduce the need for inner training from new random initializations.
> >
> > 6. > What happens if we scale the meta-gradient of BPTT with the norm of the gradient during training? The authors argue training instability of BPTT based on the larger gradient norm. Then the most straightforward way to mitigate the issue would be gradient clipping or scaling the gradient.
> >
> > We would re-iterate that in Figure 3, we demonstrate the notable instability of meta-gradients through the gradient norm. As suggested by the reviewer, we conduct an ablation study on controlling the gradient norm of BPTT. Results in Appendix C.3 shows only marginal improvements.
> >
> > Our foundational approach has already incorporated gradient clipping to manage extreme gradient norm values, employing a standard exponential moving average (EMA) with a 0.9 decay rate and capping the gradient norm at twice the adaptive norm.
> >
> > To further stabilize the gradient norm, we explored two additional methods: 1) BPTT-Gradient Clipping, limiting the gradient norm to no more than 1.05 times the adaptive norm, and 2) BPTT-Normalized Gradient, ensuring a consistent gradient norm of 1 throughout training. However, as Figure 12 illustrates, these methods achieved only marginal enhancements over the basic BPTT approach. Their performance trails behind RaT-BPTT, with a threefold increase in optimization time due to extended backpropagation.
> >
> > These findings highlight challenges such as deviation from the kernel regime, variance from extensive unrolling and pronounced non-convexity, contributing to gradient instability, as evidenced by fluctuating gradient norms. Addressing these issues solely by adjusting gradient norms proves insufficient.
> >
> > Please also see our discussion with Reviewer dNLL where we use the normalized variance metric to demonstrate the notable instability. The instability is not just in the overall norm, but also across different samples and different steps.
> >
> >
> > 7. > What happens if we gradually increase the total number of unrolling steps for BPTT instead of using fixed $T$?
> >
> > As suggested by reviewer, we examine limiting the maximum Hessian matrices in Eqn 2 by gradually extending the unrolling length $T$ in BPTT. In Figure 12 in Appendix C.3, the BPTT-Increasing Windows variant, which linearly scales $T$ from 10 to 180, underperforms both R-BPTT and standard BPTT. This again underlines the complexity within the inner loop, deviating significantly from the kernel regime and emphasizing the importance of managing the unrolling window size.

---

> > > ### Author Response · Authors · 2023-11-19
> > > **Continue on References**
> > >
> > > [1] Maclaurin, Dougal, David Duvenaud, and Ryan Adams. "Gradient-based hyperparameter optimization through reversible learning." International conference on machine learning. PMLR, 2015.
> > >
> > > [2] Sachdeva, Noveen, et al. "Farzi Data: Autoregressive Data Distillation." arXiv preprint arXiv:2310.09983 (2023).
> > >
> > > [3] Zhang, Lei, et al. "Accelerating dataset distillation via model augmentation." Proceedings of the IEEE/CVF Conference on Computer Vision and Pattern Recognition. 2023.
> > >
> > > [4] Deng, Zhiwei, and Olga Russakovsky. "Remember the past: Distilling datasets into addressable memories for neural networks." Advances in Neural Information Processing Systems 35 (2022): 34391-34404.
> > >
> > > [5] Kim, Jang-Hyun, et al. "Dataset condensation via efficient synthetic-data parameterization." International Conference on Machine Learning. PMLR, 2022.
> > >
> > > [6] Liu, Yanqing, et al. "DREAM: Efficient Dataset Distillation by Representative Matching." arXiv preprint arXiv:2302.14416 (2023).
> > >
> > > [7] Cui, Justin, et al. "Scaling up dataset distillation to imagenet-1k with constant memory." International Conference on Machine Learning. PMLR, 2023.
> > >
> > > [8] Yin, Zeyuan, Eric Xing, and Zhiqiang Shen. "Squeeze, Recover and Relabel: Dataset Condensation at ImageNet Scale From A New Perspective." arXiv preprint arXiv:2306.13092 (2023).

---

> ### Comment · Reviewer_wwkm · 2023-11-21
> **Still not convincing about its scalability**
>
> - I am not still convinced that the proposed method scales to large dataset because it still requires backpropagating through truncated inner trajectory. I am not sure the technique from [1] is applicable to the proposed method since [1] requires the inner optimization procedure "invertible". In general, Reverse Mode Differentiation requires large space complexity [2]. I am not familiar with [3], hard to say whether it does really save the memory. I highly recommend authors empirically show that some of those technique reduce the memory without performance degradation so that the proposed method scales to large datasets.
>
> - I do not think FRePO is applicable. Although we sample a feature extractor from a model pool, we need to optimize a linear classifier for inner optimization step.
>
> - Based on this, I keep my initial score as 5.
> # References
>
> [1] Maclaurin, Dougal, David Duvenaud, and Ryan Adams. "Gradient-based hyperparameter optimization through reversible learning." International conference on machine learning. PMLR, 2015.
>
> [2] Franceschi, Luca, et al. "Forward and reverse gradient-based hyperparameter optimization." International Conference on Machine Learning. PMLR, 2017.
>
> [3]  Sachdeva, Noveen, et al. "Farzi Data: Autoregressive Data Distillation." arXiv preprint arXiv:2310.09983 (2023).

---

> ### Author Response · Authors · 2023-11-22
>
> We wish to re-emphasize that within the Dataset Distillation community, Tiny-ImageNet is recognized as one of the largest datasets commonly utilized. Our review of over 20 papers reveals that only two recent studies have employed ImageNet-1K. While we are open to including results on ImageNet, the time constraints of the rebuttal phase preclude this addition. However, we would like to reinforce our confidence in our method's scalability with the following arguments, substantiated by relevant references.
>
> ## Scaling of our method
>
> Contrary to the concerns raised, we disagree with the notion that 'requiring backpropagation through a truncated inner trajectory' inherently leads to scalability issues with larger datasets. It is unrealistic to expect a method to perform equally across all datasets with identical resource allocation. A critical factor is the rate at which resource requirements increase with dataset size. When comparing ImageNet to smaller datasets such as CIFAR10, CUB-200, and TinyImageNet, we note three main areas of increase: the number of classes, the data volume per class, and the network size.
>
> • Our findings with CUB-200 and CIFAR-100 demonstrate that our method does not require additional unrolling for a larger number of classes.
>
> • An increase in data volume linearly increases training time, affecting only the number of steps per epoch. This is a natural consequence of fixed Image Per Class number, akin to the increase in epochs for standard training.
>
> • Our results with TinyImageNet (using ConvNet 4) and our generalization studies on architectures like ResNet-18, VGG-11, and AlexNet indicate that our method can adapt to these larger networks without extra unrolling steps. The memory increase is proportional to the network size, an expected and manageable outcome.
>
> Based on these observations, we conclude that our method's scaling to larger datasets is feasible and does not inherently suffer from the fact of using backpropagation through certain trajectories.
>
> ## Reducing the memory consumption
>
> Further, we propose strategies to reduce memory consumption beyond natural scaling expectations.
>
> We agree that reversing the inner optimization "requires the inner optimization procedure to be "invertible"". We note that both SGD and Adam satisfy this criterion. Given the final state and velocity (momentums), we could revert to a previous state's weights. Recalculating the gradient and then subtracting it from the velocity or momentum reverts the previous velocity (momentums). With infinite precision, such inversion is exact, as detailed in Algorithm 2 [1]. We refer the reviewer to check out a blog post at [danielewworrall.github.io/blog/2020/12/reversible-optimisers/]. [2] implements similar reversing methods for dataset distillation. While numerical issues, such as loss of precision during backward propagation, are a potential concern, the methodologies outlined in lines 9 to 13 of Algorithm [1] are designed to mitigate this. Furthermore, [3] observes significant potential for gradient reuse, suggesting the accumulation of the overall gradient during the forward pass. This approach results in an equivalent, exact method that maintains constant memory usage, and has been successfully applied to MTT, which similarly involves backpropagation through inner trajectories. Notably, this technique has led to enhanced performance.
>
> Regarding the reviewer's query about [2] "hard to say whether it really saves memory", we direct attention to the fourth figure in [2], which clearly illustrates successful memory consumption reduction. Could the reviewer kindly clarify further concerns?
>
> ## Using ideas from FRePO
>
> > Similar to FrePO, we may keep a pool of parameter checkpoints to further optimize our method. This strategy would reduce the need for inner training from new random initializations.
>
> Let us clarify again on how we could use the idea from FRePO. Instead of using random initializaions for every inner unrolling, we could choose one model randomly from the model pool and perform unroll and backward on it. This adjustment could save the forward pass time. This modification affects only lines 5 and 6 of our algorithm and the subsequent optimization and backward stages remain unchanged. This change does not conflict with "Although we sample a feature extractor from a model pool, we need to optimize a linear classifier for inner optimization step".
>
> ## References
>
> [1] Maclaurin, Dougal, David Duvenaud, and Ryan Adams. "Gradient-based hyperparameter optimization through reversible learning." International conference on machine learning. PMLR, 2015.
>
> [2] Sachdeva, Noveen, et al. "Farzi Data: Autoregressive Data Distillation." arXiv preprint arXiv:2310.09983 (2023).
>
> [3] Cazenavette, George, et al. "Dataset distillation by matching training trajectories." Proceedings of the IEEE/CVF Conference on Computer Vision and Pattern Recognition. 2022.

---

> > ### Comment · Reviewer_wwkm · 2023-11-23
> > **Reply to Authors**
> >
> > Thank you for further clarification. I could not go through the reply in detail. I will come back and read it carefully soon. Thanks

---

> > > ### Comment · Reviewer_wwkm · 2023-12-01
> > > **Update my score**
> > >
> > > I've gone through the author's responses. The authors addressed my concerns and I am happy to raise the score to 6. As Reviewer dNLL pointed it out, the current draft is 10 page and exceeds the 9 page limit. I defer the decision on whether this is acceptable to the AC.

---

### Official Review · Reviewer_dNLL · 2023-11-02

**Soundness:** 3 good
**Presentation:** 3 good
**Contribution:** 3 good
**Rating:** 8
**Confidence:** 5

**Summary:**

The paper proposes three key changes to the meta-matching framework of data distillation: (1) randomly truncated backpropagation through time (BPTT); (2) boosting in data distillation; and (3) using Adam optimizer in the inner-loop (although the authors don’t state this as a primary difference). These changes lead to SoTA results on numerous image classification datasets, especially compared to the naive meta-matching algorithm [1].

[1] Zhiwei Deng and Olga Russakovsky. Remember the past: Distilling datasets into addressable memories for neural networks.

**Strengths:**

- Thorough empirical evaluation, and SoTA performance on multiple datasets and IPC settings.
- Improved efficiency for meta-gradient computation compared to full BPTT [1] in terms of both memory & time.
- I really like the brief hardness-stratified analysis in Figure 10.

**Weaknesses:**

- Using meta-gradient norm as an indicator for stability (more in questions).
- Boosting is optimization-agnostic procedure however it’s only tested with RaT-BPTT (more in questions).
- Section 5 (boosting) is not well-discussed with key details shifted to the appendix. I would suggest keeping a subset of results from Figures 7-10 (and moving others to the appendix), but with all details for those experiments complete in the main-text.

**Questions:**

- (Figure 3) The change in gradient norm with more steps is not a good indicator of training stability. I would suggest looking at either the variance of these gradients [1], or the eigenvalues of the hessian matrix (a.k.a. sharpness) [2] as better aligned indicators.
- Do you expect the boosting idea to work with techniques other than RaT-BPTT, e.g., DSA [3], MTT [4]?
- One subtle change in RaT-BPTT is the usage of Adam optimizer in the inner-loop. Notably, all existing distillation techniques use SGD. Is this the main reason for improvement compared to other techniques? How does RaT-BPTT (SGD) compare with RaT-BPTT (Adam)? I would suggest referring to [5] for a better understanding of using Adam in data distillation.

I'd be happy to consider increasing my overall rating if some of these questions are addressed in the rebuttal.

Other comments and suggestions (not used in deciding my overall rating):
- Please include full-data performance in Table 1.
- The meta-gradient of SGD in the inner-loop can be efficiently computed [6]. Please mention it in the full-text.
- It would be great to include an analysis of the effect of varying $\beta$ in Boost-DD on downstream generalization.

[1] Faghri, Fartash, et al. "A study of gradient variance in deep learning." arXiv preprint arXiv:2007.04532 (2020).

[2] Cohen, Jeremy M., et al. "Gradient descent on neural networks typically occurs at the edge of stability." arXiv preprint arXiv:2103.00065 (2021).

[3] Bo Zhao and Hakan Bilen. Dataset condensation with differentiable siamese augmentation. In Proceedings of the International Conference on Machine Learning (ICML), pp. 12674–12685, 2021b.

[4] George Cazenavette, Tongzhou Wang, Antonio Torralba, Alexei A. Efros, and Jun-Yan Zhu. Dataset distillation by matching training trajectories. In Proceedings of the IEEE/CVF Conference on Computer Vision and Pattern Recognition (CVPR), pp. 4750–4759, 2022.

[5] Sachdeva, Noveen, et al. "Farzi Data: Autoregressive Data Distillation." arXiv preprint arXiv:2310.09983 (2023).

[6] Maclaurin, Dougal, David Duvenaud, and Ryan Adams. "Gradient-based hyperparameter optimization through reversible learning." International conference on machine learning. PMLR, 2015.

---

> ### Author Response · Authors · 2023-11-15
> **Requiring Clarifications**
>
> Thank you for your valuable suggestions. We are currently in the process of conducting the experiments as recommended.
>
> In the meantime, could you please provide further clarification on your suggestion '*It would be great to include an analysis of the effect of varying $\beta$ in Boost-DD on downstream generalization*'? What does '*downstream generalization*' mean here? We want to ensure we fully understand your feedback to effectively address it.

---

> > ### Comment · Reviewer_dNLL · 2023-11-15
> > **Clarification**
> >
> > By varying the effect of $\beta$, I meant that how does the accuracy change when e.g. training IPC 50 on CIFAR-10 but with varying values of $\beta$. So the final plot would have on x-axis as $\beta$ and the y-axis would be the accuracy of models trained on data generated using that $\beta$. Does that make sense?

---

> ### Author Response · Authors · 2023-11-19
>
> We thank the reviewer for a host of very valuable comments and suggestions and for the generally positive reception of our paper. We have tried to address most of your requests, and hope to incite you to raise your score if you like our responses.
>
> 1. > Using meta-gradient norm as an indicator for stability (more in questions). (Figure 3) The change in gradient norm with more steps is not a good indicator of training stability. I would suggest looking at either the variance of these gradients [1], or the eigenvalues of the hessian matrix (a.k.a. sharpness) [2] as better aligned indicators.
>
> Thank you for giving this suggestion! In Figure 3, we investigated the stability of meta-gradients using gradient norms as a metric, predicated on the notion that stable and efficient learning should manifest as consistent and decreasing gradient norms throughout training. Following your suggestion, we now introduce another metric for evaluating gradient stability: the normalized gradient variance, in line with the methodology proposed in [1]. Each variance value reflects the instability across the batch samples, and the values across time steps reflects the instability across training steps.
>
> To calculate this metric, we compute the average variance of all gradient entries using a set of 100 samples from the evaluation batch. Given the different scales in gradient norms across different methods, we normalize this variance against the square of the norm. This normalization yields a more consistent metric, termed the normalized variance. Employing the same experimental setup as in Figure 3, we present the results in Figure 11. It shows that RaT-BPTT not only maintains lower variance at each training step but also demonstrates more consistent variance trajectories over the course of training. These findings, in conjunction with the earlier results, collectively offer a comprehensive view of the argued training instability. We have added the discussion in a new section (C.3) in the Appendix, as we feel it further strengthens our analysis and thank this reviewer again for the very valuable suggestion!
>
> We do not employ the Hessian analysis due to the prevailing uncertainty in the field regarding the learning dynamics at the edge of stability, particularly when using the Adam optimizer with moderate and small batch sizes. This aspect of optimization remains a topic of ongoing research, as current results, like those presented in [2, 7], are predominantly confined to scenarios involving full batch training. Given this limitation, it becomes challenging to establish a definitive standard for assessing gradient stability from the Hessian perspective. The variance of gradients instead stands as a more interpretable metric and our results again validate the improvements of RaT-BPTT.
>
> 2. > Boosting is optimization-agnostic procedure however it’s only tested with RaT-BPTT (more in questions). Do you expect the boosting idea to work with techniques other than RaT-BPTT, e.g., DSA [3], MTT [4]?
>
> Yes, thank you for asking this question, prompting us to add what we hope is a valuable proof of concept. Indeed, we believe that the Boosting framework will work with other methods. As a proof of principle, and choosing a representative method, we implement strong boosting (with $\beta = 0$) for MTT from IPC 10 to IPC 50 in steps of IPC 10. The final generalization accuracy is 71.4%. The accuracy improves rapidly throughout the boosting and the final performance is marginally lower (by 0.2%) than the direct distillation of IPC50 with MTT, which stands at 71.6%. We have added the discussion in a new section (C.6) in the Appendix.

---

> > ### Author Response · Authors · 2023-11-19
> > **Continue**
> >
> > 3. > One subtle change in RaT-BPTT is the usage of Adam optimizer in the inner-loop. Notably, all existing distillation techniques use SGD. Is this the main reason for improvement compared to other techniques? How does RaT-BPTT (SGD) compare with RaT-BPTT (Adam)? I would suggest referring to [5] for a better understanding of using Adam in data distillation.
> >
> > We have opted for Adam instead of SGD to simplify the tuning process for the inner loop. This decision was based on the ability to use a common learning rate without requiring decay in the inner loop.
> >
> > As the reviewer suggested, we have implemented RaT-BPTT (SGD) using SGD with learning rate 0.01 and learning rate decays at [120, 200] by 0.2. For IPC10 on CIFAR10, RaT-BPTT (SGD) achieves a 69.0% accuracy (std 0.3%), while RaT-BPTT (Adam) results in a slightly higher accuracy of 69.4% (std 0.4%). Thus, RaT-BPTT (SGD) also outperforms previous methods in this setting by a large margin. It is crucial to note that our improvement is attributed to factors beyond merely employing Adam in the inner loop. We are grateful to the reviewer for prompting this ablation study which, we believe, gives further credence to the RaT-BPP method. Given that RaT-BPP outperforms the benchmarks even when using SGD, we hope that it is not necessary to run some of the benchmark methods with Adam. In fact, we tried to combine MTT with Adam. However, either using Adam in both the teacher and student models or only in the student yields subpar performance, different from the observations in [5]. This discrepancy may be attributed to inherent differences between language and image data processing – with Adam being the predominant optimizer in NLP, while SGD maintains its prevalence in image classification tasks. As [5] unfortunately did not provide detailed configurations for their experiments, we are open to exploring these aspects in greater depth after communicating with their authors.
> >
> > It is also noteworthy to point out that we are not the first to use Adam for the inner loop during training. [8] also uses Adam for their linearized inner loop. Some other papers [8, 9] have also adopted Adam for the linear loop during evaluations. We suspect that whenever Adam was an option, the benchmarking papers probably tried it without significant improvements.
> >
> > We hope this response on Adam versus SGD is satisfactory to the reviewer and thank you again for this valuable suggestion.
> >
> >
> >
> >
> > 4. > Please include full-data performance in Table 1.
> >
> > We have updated Table 1 in the revised paper as per your suggestion, and hope these numbers are helpful.
> >
> >
> > 5. > The meta-gradient of SGD in the inner-loop can be efficiently computed [6]. Please mention it in the full-text.
> >
> > Thank you for highlighting the significance of references [5] and [6]. These works indeed open new avenues for accelerating RaT-BPTT. We have now included this discussion to acknowledge their contributions and point out future directions in Section 6.
> >
> > 6. >It would be great to include an analysis of the effect of varying $\beta$ in Boost-DD on downstream generalization.
> >
> > As for the Reviewer's suggestion to give a comprehensive study of various boosting strengths $\beta$: we will include that in the final version (in addition to the current $\beta=0$ and $\beta=0.1$). Our preliminary studies indeed show a graceful decline of accuracy on the first block with increasing $\beta$, as one can expect from interpolating through $\beta=0, 0.1$. Unfortunately, running a comprehensive grid is time consuming, given the iterative nature of boosting from smaller to larger blocks. What we can conclude, given that the final accuracy of a strongly iteratively boosted IPC50 dataset is very close to the accuracy when optimizing IPC50 from scratch, is that the final accuracy for IPC50 for all $\beta$ will be very similar - but filling in the gaps for smaller IPC will take a little more time.
> >
> >
> > [7] Cohen, Jeremy M., et al. "Adaptive gradient methods at the edge of stability." arXiv preprint arXiv:2207.14484 (2022).
> >
> > [8] Loo, Noel, et al. "Dataset Distillation with Convexified Implicit Gradients." arXiv preprint arXiv:2302.06755 (2023).
> >
> > [9] Zhou, Yongchao, Ehsan Nezhadarya, and Jimmy Ba. "Dataset distillation using neural feature regression." Advances in Neural Information Processing Systems 35 (2022): 9813-9827.

---

> > > ### Comment · Reviewer_dNLL · 2023-11-23
> > > **Response to authors**
> > >
> > > Thanks for the comprehensive response to my questions, and including a variety of new experiments. Overall, I'm happy with the paper and have increased my rating to 8 accordingly.
> > >
> > > One thing I leave for the Area Chair to decide is that the updated PDF now contains 10 pages which is more than the max page limit.

---

### Official Review · Reviewer_Qtcx · 2023-11-03

**Soundness:** 2 fair
**Presentation:** 3 good
**Contribution:** 2 fair
**Rating:** 5
**Confidence:** 2

**Summary:**

This paper proposes Random Truncated BPTT (RaT-BPTT) for solving the bi-level optimization problem in dataset distillation. RaT-BPTT incorporates a truncation coupled with a random window, effectively stabilizing the gradients and speeding up the optimization while covering long dependencies. Empirical results show that such simple method outperforms existing methods for dataset distillation application.

**Strengths:**

- The idea is simple and easily applicable in practice
- The empirical results are positive

**Weaknesses:**

- No theoretical results supporting the empirical results. Analyzing the bi-level optimization problem might be difficult in general, but it is better to find some simple setting where theory can explain when/why the proposed method outperforms existing methods.

**Questions:**

Is the proposed algorithm applicable for general bi-level optimization problems, not only for the dataset distillation application? If so, it would be great to add some discussions on what other applications using bi-level optimization (which currently use BPTT) one can use RaT-BPTT instead.

---

> ### Author Response · Authors · 2023-11-19
>
> We thank the reviewer for their time and valuable feedback. We hope the following responses address your questions.
>
> 1. > No theoretical results supporting the empirical results. Analyzing the bi-level optimization problem might be difficult in general, but it is better to find some simple setting where theory can explain when/why the proposed method outperforms existing methods.
>
> In numerous analyses of bi-level optimization, a common assumption is the convexity of the inner problem, with a focus on convergence to stationary points. However, in scenarios where the inner problem exhibits local strong convexity, along with other favorable conditions, it is unreasonable to expect that the convergence rate of RaT-BPTT would be faster in orders than traditional T-BPTT. On the other hand, in non-convex settings, which are typical of deep learning problems like ours and where RaT-BPTT demonstrates notable empirical performance, providing theoretical proofs of effectiveness becomes exceedingly challenging.
>
> Nonetheless, in Section 3 we have aimed to provide a heuristic to justify the increased performance of RaT-BPTT, particularly from a gradient perspective. The inherent non-convexity in the inner loop necessitates long unrolling to adequately capture long term dependencies. This requirement, however, introduces significant computational challenges and leads to gradient instability due to the compounding effects of non-convex Hessian matrix products. RaT-BPTT is a natural solution to all three problems, by limiting the window to control stability and reduce computational burden, and by moving the window to cover the entire unrolling. Both gradient inspections and empirical performance support such heuristics.
>
> To provide a different angle for theoretical justification, we lay out another hypothesis related to kernel approximations of neural nets (like the NTK). This hypothesis attributes the success of RaT-BPTT, in comparison to both BPTT and T-BPTT, to the separation between feature learning and kernel learning. Our inner loop is training a neural network from scratch. According to [1], there is a significant rotation in the empirical kernel at the onset of training, which then gradually stabilizes. This observation suggests a dominant role of feature learning initially, followed by kernel learning.
>
> From this perspective, for large unrolling $T$, T-BPTT is akin to optimizing kernel learning from $\theta_{T-M}$. The initialization distribution $p_{\theta_{T-M}}$ is only loosely related to the data. Therefore, T-BPTT only optimizes data that favors good kernel learning, without any assurance of effective feature learning optimization. In contrast, RaT-BPTT, with its sliding window, explicitly focuses on also optimizing the feature learning aspect, which is crucial, especially in the initial stages of training.
>
> Previous works [2] have established provable benefits of feature learning with specifically tailored noises. We find it promising to adopt such constructions into bi-level optimziation problems to show different features captured by feature learning and kernel learning. However, we believe that such a theoretical guarantee is beyond the scope of our empirical study and likely requires an in-depth study of kernel evolution during unrolling, which likely will have empirical components as well.
>
> 2. > Is the proposed algorithm applicable for general bi-level optimization problems, not only for the dataset distillation application? If so, it would be great to add some discussions on what other applications using bi-level optimization (which currently use BPTT) one can use RaT-BPTT instead.
>
> Yes! Thanks for pointing it out. As suggested, we have added a new paragraph in Section 6. This addition discusses the broader implications of our work in bilevel optimizations, specifically highlighting its potential in meta-learning for hyperparameters, MAML-like problems, and Poisoning Attacks.
>
> [1] Fort, Stanislav, et al. "Deep learning versus kernel learning: an empirical study of loss landscape geometry and the time evolution of the neural tangent kernel." Advances in Neural Information Processing Systems 33 (2020): 5850-5861.
>
> [2] Karp, Stefani, et al. "Local signal adaptivity: Provable feature learning in neural networks beyond kernels." Advances in Neural Information Processing Systems 34 (2021): 24883-24897.

---

### Author Response · Authors · 2023-11-19
**General Response**

We would like to draw the attention of all reviewers to the following points:

1. Update of Figure 3:

We apologize for confusion that might have been caused by a slightly erroneous Figure 3 in the original submission. In there, the line representing RaT-BPTT was inaccurate. We have corrected it now. Please note that the new Figure more clearly shows that RaT-BPTT enjoys a more stable gradient with respect to the norm.

Following valuable suggestions of Reviewer 2 (dNLL) we have further added a similar figure studying gradient *variance* (Figure 11). The results for variance align with those for the norm.

2. Additions to the paper manuscript:

We have added several new Appendices. C.3 and C.4. give a variety of new ablation studies in response to the reviewers' questions and valuable suggestions, to further show the merits of RaT-BPPT. C.5. now shows studies with more architectures, also demonstrating that our distilled dataset is very transferable across architectures.  C.6 shows an ablation of Adam versus SGD for RaT-BPP to clarify that the inner optimizer is *not* the source of improvement. C.7 discusses the efficiency of our method. C.8. shows that our BoostDD framework also extends to other dataset distillation methods, as shown for MTT (Matching Trajectories). We make short references to them at the end of Section 3, 4, and 5 now. We have also expanded the discussion (Section 6) to highlight a few more points brought up by the reviewers. **All the updates are marked in Cyan in the PDF.**

---

### Meta-Review · Area_Chair_ruHg · 2023-12-02

**Metareview:**

This paper received several domain expert reviews and pointed out both merits and limitations. The scalability issue is not well addressed in the current version. I encourage authors could add some more convincing experiments in large-scale datasets.

More importantly, one review pointed out this submission exceeds the upper paper limit of 9 pages. I checked the call for papers. it says: "There will be a strict upper limit of 9 pages for the main text of the submission, with unlimited additional pages for citations. This page limit applies to both the initial and final camera-ready version."
This is a *strict* upper limit and it applies to all versions without any exception.

Due to the above reasons, I have to reject this paper.

**Justification For Why Not Higher Score:**

Violation of paper submission guidelines.

**Justification For Why Not Lower Score:**

NA

---

### Decision · Program_Chairs · 2024-01-16

**Decision:**

Accept (poster)

**Comment:**

The paper was originally rejected because its post-rebuttal version was slightly over 9 pages long. The authors correctly pointed out that the CFP didn't specify length limits for intermediate versions of papers and that there were accepted papers whose post-rebuttal versions also exceeded the page limit. The PC chairs deliberated over the merits of the paper, concluded that the paper would have been accepted if the slight length overflow hadn't been a consideration, and decided to accept the paper.